# A large-strain and ultrahigh energy density dielectric elastomer for fast moving soft robot

Wenwen Feng[1], Lin Sun[1], Zhekai Jin[1], Lili Chen[1], Yuncong Liu[1], Hao Xu[1] & Chao Wang [1] ✉

Dielectric elastomer actuators (DEAs) with large actuation strain and high energy density are highly desirable for actuating soft robots. However, DEAs usually require high driving electric fields (>100 MV m$^{-1}$) to achieve high performances due to the low dielectric constant and high stiffness of dielectric elastomers (DEs). Here, we introduce polar fluorinated groups and nanodomains aggregated by long alkyl side chains into DE design, simultaneously endowing DE with a high dielectric constant and desirable modulus. Our DE exhibits a maximum area strain of 253% at a low driving electric field of 46 MV m$^{-1}$. Notably, it achieves an ultrahigh specific energy of 225 J kg$^{-1}$ at only 40 MV m$^{-1}$, around 6 times higher than natural muscle and twice higher than the state-of-the-art DE. Using our DE, soft robots reach an ultrafast running speed of 20.6 BL s$^{-1}$, 60 times higher than that of commercial VHB 4910, representing the fastest DEA-driven soft robots ever reported.

Dielectric elastomers (DEs) are electric-response materials that deform significantly by electrostatic forces under external electric fields[1]. Dielectric elastomer actuators (DEAs), a device of DEs coated with compliant electrodes, possess many advantages for actuating soft robots, such as large actuation strain, excellent energy output performance, fast actuation response, and high bandwidth[2–6]. Among them, the large actuation strain and high energy density are key to actuating high-performance mobile soft robots for fast moving[7–9], high jumping[10,11], and even lift-off[12]. In the available DEs, commercial very high bond (VHB) acrylic elastomers (3 M) are commonly used for soft robots thanks to their satisfying actuation performance. However, the low dielectric constant and high stiffness of polymer networks make VHB-based DEAs require a high driving electric field (typically > 100 MV m$^{-1}$) to achieve large actuation strain and high energy density, which severely limits the widespread application of DEAs[2,13,14]. Extensive efforts have been devoted to improving the actuation performance of DEs at low driving electric fields, including bottlebrush elastomer[15,16], poly[styrene-b-(ethylene-co-butylene)-b-styrene] (SEBS) triblock copolymer mixed with oil[17], pentablock copolymer[18], and polyacrylate with optimized crosslinking network[19]. Although these

materials advance in mechanical flexibility, they have not yet achieved large actuation strain (>200%) and high energy density (>150 J kg$^{-1}$) simultaneously, due to their low dielectric constant. As a result, it remains challenging to develop high-performance DEs with large actuation strain and high energy density under low electric fields.

Here, we develop a high-performance polar fluorinated polyacrylate DE with a high dielectric constant (10.23, 1 kHz) and desirable Young's modulus (~0.09 MPa). Meanwhile, we employ nanodomains aggregated by long alkyl side chains as physical crosslinkers to adjust the mechanical properties and enhance the electromechanical properties of DE (Fig. 1a, b). This strategy successfully endows the DE with large actuation strain and high energy density under low electric fields. As shown in Fig. 1c, our DE achieves a maximum actuation area strain of 253% at only 46 MV m$^{-1}$, which is superior to most DEs developed in recent years[2,14,18–25]. Notably, our DE exhibits an ultrahigh specific energy (mass energy density) of 225 J kg$^{-1}$ and a high specific power (mass power density) of 2245 W kg$^{-1}$ at 40 MV m$^{-1}$ and 5 Hz, exceeding that of natural muscle almost 6 times and outperforming all of the reported DEs (Fig. 1d)[5,6,9,12,20,26–32]. In addition, our DE can be stably actuated above a high specific energy of 150 J kg$^{-1}$ for 10,000 cycles.

[1]Key Lab of Organic Optoelectronics & Molecular Engineering, Department of Chemistry, Tsinghua University, 100084 Beijing, China.
✉e-mail: chaowangthu@mail.tsinghua.edu.cn

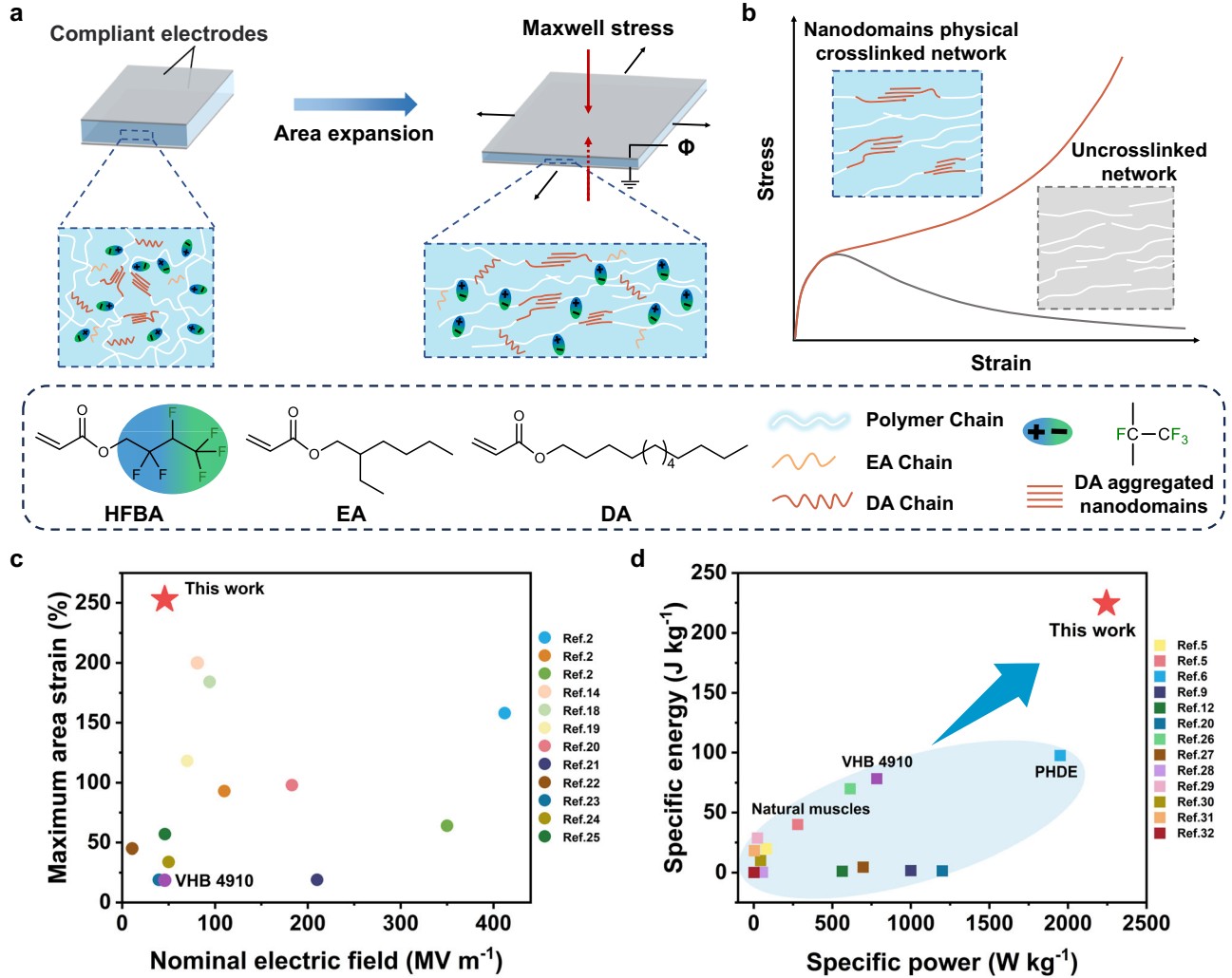

**Fig. 1 | Polymer design and high-performance actuation. a** Schematic illustration of actuation mechanism and chemical structures of our high-performance DE. **b** Variation of the stress-strain curves of DA-aggregated nanodomains physical crosslinked network and uncrosslinked network. **c** Ashby plot summarizing maximum actuation area strains to electric fields of this work and other pre-stretched and stable DEs. **d** Ashby plot summarizing specific energy to specific power of this work and other DEs. VHB 4910 represented the measured value of VHB 4910 in this work.

With excellent energy output performance, our artificial muscles can lift a 1.5 kg bucket. We further fabricate an ultrafast moving soft robot driven by DEA with a minimum energy structure to demonstrate the excellent actuation performance of our DE. Based on our high-performance DE, the soft robot is the fastest DEA-driven soft robot up to now[7–11,33–41], achieving a running speed of up to 20.6 BL s[−1] (body lengths per second), 60 times higher than that of VHB 4910 and even comparable to the speed of cheetah[42]. Moreover, our soft robot can climb slopes up to 45° and carry a load of around 17 times its own weight.

## Results

### Polymer design

To develop a high-performance DE under low electric fields, we design a random copolymer of 2,2,3,4,4,4-hexafluorobutyl acrylate (HFBA), 2-ethylhexyl acrylate (EA) and dodecyl acrylate (DA) (Fig. 1a). The transparent dielectric elastomer was synthesized by mixing the three comonomers in one-step UV photopolymerization (Supplementary Fig. 1). In this strategy, the rich and highly polar CF$_3$ groups in HFBA segments provide high dielectric constant. EA with large steric hindrance side chains is selected as comonomers to lower the Young's modulus of the copolymer (Supplementary Fig. 2). DA is able to participate in crystallization due to the packing of long alkyl side

chains[43,44] (Supplementary Fig. 3). Therefore, in the copolymer, the long alkyl side chains in DA can effectively aggregate to form nanodomains and serve as dynamic physical crosslinkers to enhance the elasticity and achieve strain-hardening behavior, which are essential for excellent actuation performance[1,3]. With the combination of high dielectric constants and desirable mechanical properties, our dielectric elastomer is expected to display large actuation strain and high energy density under low electric fields.

### Mechanical properties and dielectric properties

To tune the mechanical and dielectric properties of our DE, the molar ratios of HFBA, EA, and DA were systematically changed (Supplementary Table 1). As shown in Fig. 2a, the polymerized HFAB (PF) was stiff, so EA was added to soften the copolymer (PFE). Due to the poor elasticity and low dielectric constant of EA, excessive addition of EA will decrease the elasticity and dielectric constant of the copolymer (Supplementary Figs. 4 and 5). Therefore, we fixed the molar ratio of EA relative to HFBA at 10%. Unfortunately, PFE showed obvious yield behavior, which was not suitable for DEs applications (Supplementary Fig. 6). However, by adding only a small amount of DA to PFE, the stress-strain curve of the copolymer exhibited strain-hardening behavior. The copolymers of HFBA, EA, and DA were named PFED-x, where x was the molar ratio of DA relative to HFBA (Supplementary Table 1).

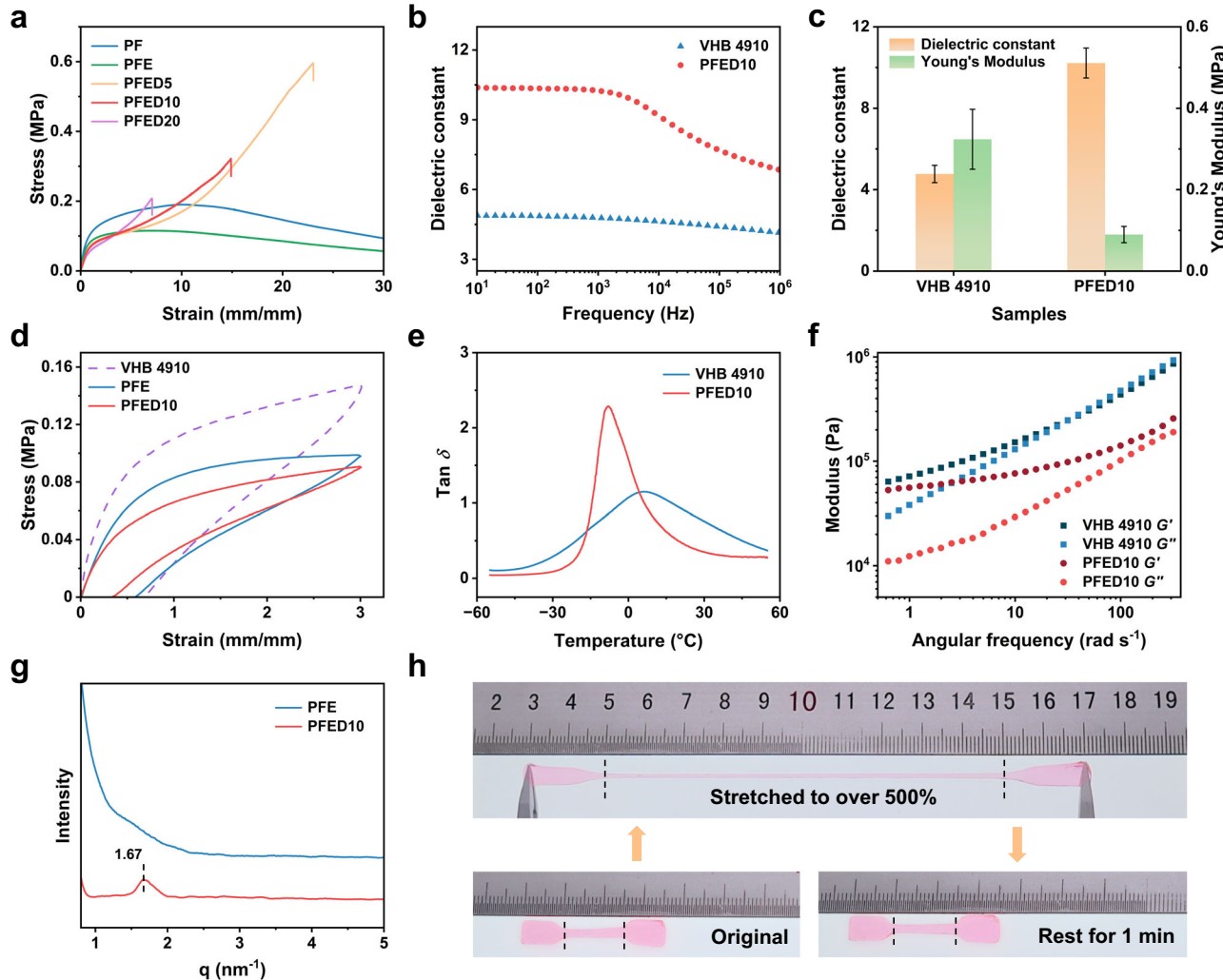

**Fig. 2 | Mechanical properties and dielectric properties. a** Stress-strain curves of PF, PFE, PFED5, PFED10, and PFED20. **b** Dielectric constants of VHB 4910 and PFED10 from 10 Hz to $10^6$ Hz. **c** Comparison of dielectric constant (measured at 1 kHz) and Young's modulus of VHB 4910 and PFED10. Error bars show s.d., $n = 3$. **d** Cyclic stress-strain curves of PFE, PFED10, and VHB 4910 subjected to strain at 300%. **e** Mechanical losses of VHB 4910 and PFED10 from −55 °C to 55 °C. **f** Storage modulus ($G'$) and loss modulus ($G''$) of VHB 4910 and PFED10 versus angular frequency from 0.1 to 400 rad s$^{-1}$ at 25 °C. **g** SAXD patterns of PFE and PFED10. **h** Photographs of PFED10 sample being stretched to over 500% strain and recovering in 1 min.

In the series of PFED copolymers, increasing the DA content led to lower stretchability, which indicated the higher DA content made for the formation of larger-size poly (dodecyl acrylate) (PD) nanodomains (Fig. 2a). In addition, all PFED copolymers can achieve a large actuation strain more than 150%, far exceeding that of PFE, indicating that the strain-hardening behavior is necessary to achieve large actuation performance[3,45] (Supplementary Fig. 6). In the series of PFED copolymers, PFED10 exhibited the largest actuation area strain. Therefore, considering the mechanical properties, dielectric properties, and actuation area strain, the PFED10 was selected for our study. The results of $^1$H NMR, solid-state $^{13}$C NMR, and solid-state $^{19}$F NMR characterized the chemical structure of the PFED10 (Supplementary Figs. 7–9). In Fourier transform infrared (FTIR) and Raman spectra, the disappearance of $\nu$(C=C) (from comonomers) at 1637 cm$^{-1}$ and 1643 cm$^{-1}$, respectively, indicated that the conversion of comonomers was almost 100% (Supplementary Fig. 10). In addition, the weight-average molecular weight ($M_w$) of PFED10 was 37,761 Da (Supplementary Fig. 11). The PFED10 elastomer possessed Young's modulus of 0.09 MPa (Fig. 2c), around one-third of that of commercial VHB 4910 (~0.32 MPa). Owing to the highly polar CF$_3$ groups in HFBA segments, PFED10 elastomer exhibited a high dielectric constant of 10.23 at 1 kHz,

which was much higher than that of VHB 4910 (4.77, at 1 kHz) (Fig. 2b). With the increase of frequency, the dielectric losses of PFED10 and VHB 4910 generally increased, which led to the decrease of the total electro-mechanical efficiency and actuation area strain (Supplementary Fig. 12). However, compared with dielectric losses, mechanical losses dominate the total electro-mechanical efficiency[46]. In addition, we tested the conductivity of PFED10. PFED10 was shown to be electrically insulating in the range of 10 Hz to $10^6$ Hz and the conductivity of PFED10 changed slightly from −20 °C to 100 °C indicating a wide operating temperature range (Supplementary Figs. 13 and 14).

The low viscoelasticity is critical for DEs to improve the electro-mechanical response speed and bandwidth. In PFED10, DA segments can not only achieve strain-hardening behavior but also increase the elasticity (Supplementary Fig. 4). The cyclic tensile tests were performed to demonstrate the good elasticity of PFED10 elastomer. Under different cyclic tensile strains, PFED10 always showed smaller hysteresis and residual strain than PFE and VHB 4910 (Fig. 2d and Supplementary Fig. 15) and the cyclic stress-strain curves almost overlapped for several stretching cycles (Supplementary Fig. 16). Even being stretched to over 500%, PFED10 can fully recover within 1 min (Fig. 2h). Moreover, the mechanical loss of PFED10 (tan $\delta$ = 0.41, 1 Hz,

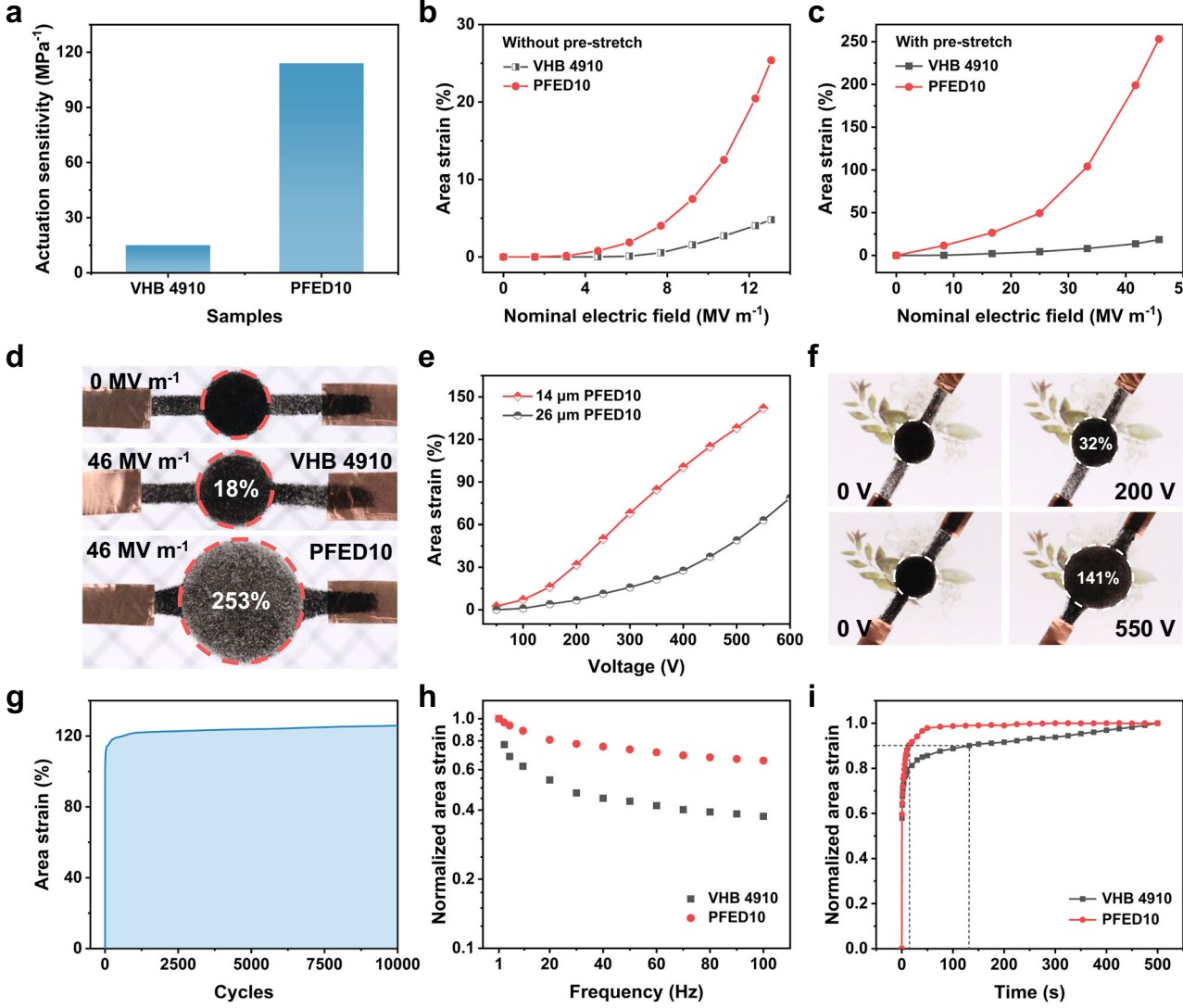

**Fig. 3 | Actuation properties. a** Comparison of actuation electromechanical sensitivity ($\beta$) of VHB 4910 and PFED10. **b** The actuation area strains of VHB 4910 and PFED10 under different electric fields without pre-stretch. **c** The actuation area strains of VHB 4910 and PFED10 under different electric fields with 275% biaxial pre-stretch. **d** Photographs comparison of the actuation area strains of VHB 4910 and PFED10 with 275% biaxial pre-stretch at 46 MV m$^{-1}$. **e** The actuation area strains of 14 μm and 26 μm PFED10 films as a function of voltage. **f** Photographs of the actuation area strains of 14 μm PFED10 film at low voltages. **g** Cyclic actuation of PFED10 at 37.5 MV m$^{-1}$ and 1 Hz for 10,000 cycles with 275% biaxial pre-stretch. **h** Frequency response of VHB 4910 and PFED10 at 37.5 MV m$^{-1}$ with 275% biaxial pre-stretch in the range of 1 Hz to 100 Hz. The actuation area strain at 1 Hz was normalized to 1. **i** Time-dependent response of VHB 4910 and PFED10 at 25 MV m$^{-1}$ with 275% biaxial pre-stretch. The actuation area strain of 500 s was normalized to 1.

25 °C) was lower than VHB 4910 (tan $\delta$ = 0.85, 1 Hz, 25 °C), also suggesting PFED10 had higher elasticity than VHB 4910 (Fig. 2e and Supplementary Fig. 17). To investigate the elasticity of PFED10 at different frequencies, we further performed rheological tests. As shown in Fig. 2f, for PFED10, the storage modulus ($G'$) exceeded the loss modulus ($G''$) over the whole angular frequency range from 0.1 to 400 rad s$^{-1}$, demonstrating PFED10 possessed excellent elasticity and high bandwidth as DE. In contrast, the $G''$ of VHB 4910 was higher than $G'$ after 40 rad s$^{-1}$. To verify the structure of PFED10, we used wide-angle X-ray diffraction (WAXD) and small-angle X-ray diffraction (SAXD) tests (Fig. 2g and Supplementary Figs. 18 and 19). The SAXD result confirmed that the strain-hardening behavior and good elasticity of PFED10 arose from the nanodomains aggregated by DA segments, which served as dynamic physical crosslinkers. Compared to PFE, the SAXD result of PFED10 showed a peak similar to that of PD[44], indicating the presence of nanodomains. The average distance $d$ between nanodomains ($d = 2\pi/q_{max}$) was approximately 3.76 nm (Fig. 2g and Supplementary Fig. 19). In addition, the PFED10 elastomer

possessed hydrophobicity and self-healing capability (Supplementary Figs. 20 and 21). The low glass transition temperature (Supplementary Fig. 22), high density of dipole-dipole interactions between CF$_3$ groups, and excellent hydrophobicity allowed the polymer chains of PFED10 to diffuse and re-entangle at the damaged locations without being affected by the aqueous environment[47], endowing PFED10 with underwater self-healing capability. Low Young's modulus, high dielectric constant, and good elasticity show that the PFED10 elastomer can have excellent actuation performance.

## Actuation properties

Electromechanical sensitivity ($\beta$), defined as the ratio of dielectric constant to Young's modulus, is an important parameter to evaluate the theoretical actuation performance of DEs. As shown in Fig. 3a, the electromechanical sensitivity of PFED10 is up to 114, which is more than 7 times higher than that of VHB 4910 ($\beta$ = 15), exceeding that of most reported DEs[19,22,48–53], suggesting excellent actuation performances of PFED10 (Supplementary Table 2). The actual actuation

performances of PFED10 and VHB 4910 were first tested without pre-stretch. PFED10 exhibited a maximum area strain of 25% at 13 MV m$^{-1}$, while VHB 4910 is only 5% at the same electric field (Fig. 3b). Further, the actuation performances were tested with 2.75 × 2.75 biaxial pre-stretch. Surprisingly, as shown in Fig. 3c, d, and Supplementary Movie 1, PFED10 achieved a maximum area strain of 253% at only 46 MV m$^{-1}$ thanks to the high actuation sensitivity, which was almost 14 times larger than that of VHB 4910 and superior to most DEs developed in recent years (Fig. 1c). More importantly, the high actuation sensitivity allows PFED10 film to be actuated at a very low voltage of 100 V. As illustrated in Fig. 3e, f, the 14 μm PFED10 film showed 7%, 32%, and 141% maximum area strain at 100 V, 200 V, and 550 V, relatively. Notably, PFED10 exhibited excellent cycle stability for reliable operation, regardless of the application of pre-stretch (Fig. 3g and Supplementary Fig. 23). As shown in Fig. 3g, the pre-stretched PFED10 was successfully actuated for 10,000 cycles under a large actuation area strain of 120% at 1 Hz.

Meanwhile, the low viscoelasticity of PFED10 endows DEA with excellent dynamic response performance, including frequency response and time response. For the frequency response, when a square wave electric field of 37.5 MV m$^{-1}$ was applied to the pre-stretched VHB 4910 film from 1 Hz to 100 Hz, the normalized actuation area strain of VHB 4910 rapidly decayed below 50% at 30 Hz. In sharp contrast, at the same test conditions, the PFED10 film kept 66% of the initial actuation area strain even at 100 Hz, showing a high bandwidth (Fig. 3h). For the time response, PFED10 took 14 s to achieve 90% of the final area strain and maintained the strain essentially constant for the subsequent time, indicating a fast response speed. However, it took 132 s for VHB 4910 to achieve 90% of the final area strain, and the strain improved clearly with the increase of actuation time, showing obvious viscoelasticity (Fig. 3i). In addition, the PFED10-based DEA has self-healing property. Upon mechanical scratches, the DEA can self-heal and maintain the basic actuation performance (Supplementary Figs. 24–26).

## Energy density and power density

Due to the high dielectric constant and large actuation area strain, PFED10 exhibits high energy density and power density at low electric fields. A pure-shear DEA was fabricated by applying a fixed pre-stretch in one plane direction and loading in the perpendicular plane direction to test the specific energy and specific power of the PFED10 and VHB 4910 (Fig. 4a). When voltage was applied, the expansion of DEA in the lateral direction was constrained, while it could freely expand in the load direction to produce the linear actuation. The blocking force was used to characterize the force output, which was measured by further constraining the deformation of the pure-shear DEA in the load direction. With a load of 60 g, the specific energy of PFED10 can reach up to 75 J kg$^{-1}$ during contraction at 20 MV m$^{-1}$ (block force ~340 mN) while VHB 4910 is only 16 J kg$^{-1}$ (Supplementary Fig. 27). Impressively, PFED10 achieved 50% linear strain and exhibited a maximum specific energy of 208 J kg$^{-1}$ with a load of 120 g at 40 MV m$^{-1}$ and 0.5 Hz (block force ~594 mN), which was almost 3 times higher than that of VHB 4910 (71 J kg$^{-1}$) (Fig. 4b, c and Supplementary Fig. 28). We further tested the frequency response of DEs' specific energy and specific power under the load of 120 g. PFED10 maintained high specific energy at low frequency (Fig. 4d). Notably, at 5 Hz, PFED10 reached an ultrahigh specific energy of 225 J kg$^{-1}$, which was almost 6 times higher than that of natural muscle (0.4–40 J kg$^{-1}$) and even twice higher than the state-of-the-art dielectric elastomers[6] (~100 J kg$^{-1}$), and the corresponding specific power was up to 2245 W kg$^{-1}$ (Fig. 4d, e, and Supplementary Movie 2). In contrast, the specific energy and specific power of VHB 4910 were 78 J kg$^{-1}$ and 784 W kg$^{-1}$, respectively, at 5 Hz. As the frequency increases, the specific energy and corresponding specific power of PFED10 remained at 43 J kg$^{-1}$ and 855 W kg$^{-1}$ at 10 Hz, and 21 J kg$^{-1}$ and 855 W kg$^{-1}$ at 20 Hz. The maximum specific energy and specific power of PFED10 surpass all the reported DEs (Fig. 1d).

Moreover, PFED10 exhibited excellent cycle stability under pure-shear linear actuation as well. PFED10 can output a high specific energy above 150 J kg$^{-1}$ and a high specific power above 1500 W kg$^{-1}$ for 10000 cycles (Supplementary Fig. 29). To scale up the energy output, we prepared two larger planar pure-shear DEAs with increased active area. Due to the ultrahigh specific energy of PFED10, the two combined planar DEAs were able to lift a 1.5 kg bucket and still achieve 35% linear actuation strain (Fig. 4f and Supplementary Movie 3).

## Ultrafast moving soft robot

With our excellent PFED10 DE, we prepared an ultrafast moving soft robot driven by DEA with a minimum energy structure (Supplementary Figs. 30 and 31). To simplify, the theoretical locomotion speed ($V$) is related to frequency ($f$) and span ($\delta$, the distance between front feet and back feet)[54], namely $V = f \times \delta$. With the increase of driving electric field, the span increased. Due to the large actuation area and high bandwidth, the PFED10-based soft robot always maintained a much larger span than that of VHB 4910 even at high frequencies (Fig. 5a). The large span and ultrahigh specific energy allowed the PFED10-based soft robot enough driving force to achieve fast moving. As a result, the running speed of the PFED10-based soft robot increased monotonically with the electric field and the speed-frequency curve displays a peak (Fig. 5b and Supplementary Fig. 32). As shown in Fig. 5b, d, and Supplementary Movie 4, the soft robot based on PFED10 exhibited the maximum speed of 20.6 BL s$^{-1}$ at 38 MV m$^{-1}$ and 30 Hz on the sawtooth-shaped substrate, which is 60 times larger than that of the soft robot based on VHB 4910 (0.33 BL s$^{-1}$). The PFED10-based soft robot demonstrated high relative speed and posture like a cheetah, making it the fastest among published DEA-driven soft robots (Fig. 5c, d, k). Besides, the PFED10-based soft robot can run fast on different substrates (Fig. 5e and Supplementary Movie 5). Benefiting from the ultrahigh specific energy, the PFED10-based soft robot possessed good load-carrying ability. It could carry a load of up to 10.6 g with a moving speed of 0.43 BL s$^{-1}$, almost 17 times its own weight (Fig. 5f, i, and Supplementary Movie 6). Meanwhile, it could climb a slope up to 45° (Fig. 5g and Supplementary Movie 7). At a slope of 20°, it could keep a moving speed as fast as 3.5 BL s$^{-1}$ (Fig. 5j, Supplementary Fig. 33, and Supplementary Movie 7). In addition, the soft robot based on PFED10 had good robustness (Supplementary Fig. 34). Due to the excellent low-voltage actuation properties of PFED10, we further prepared ultrathin films to achieve a DEA-driven soft robot with low voltage. As shown in Fig. 5h and Supplementary Movie 8, the soft robot fabricated by 26 μm PFED10 film can run at 400 V.

## Discussion

In summary, we propose a polar fluorinated polyacrylate DE with nanodomains dynamic physical crosslinked network. This strategy readily endows DE with high dielectric constant and mechanical flexibility to simultaneously achieve large actuation strain (253%) and ultrahigh specific energy (225 J kg$^{-1}$) under low electric fields, which is the highest specific energy in all reported DEs. Based on our high-performance dielectric elastomer, we further demonstrated a soft robot possessed the fastest running speed among DEA-driven soft robots (20.6 BL s$^{-1}$) and excellent load-carrying capacity and climbing ability. In addition, we achieved a 400 V driven low-voltage soft robot. We believe that our strategy promises to broaden the development of high-performance DEs under low electric fields and make it possible to apply the next generation of high-performance mobile soft robots.

## Methods
### Materials
2,2,3,4,4,4-Hexafluorobutyl acrylate (HFBA) was purchased from Shanghai Shangfu Company. 2-Ethylhexyl acrylate (EA) and dodecyl

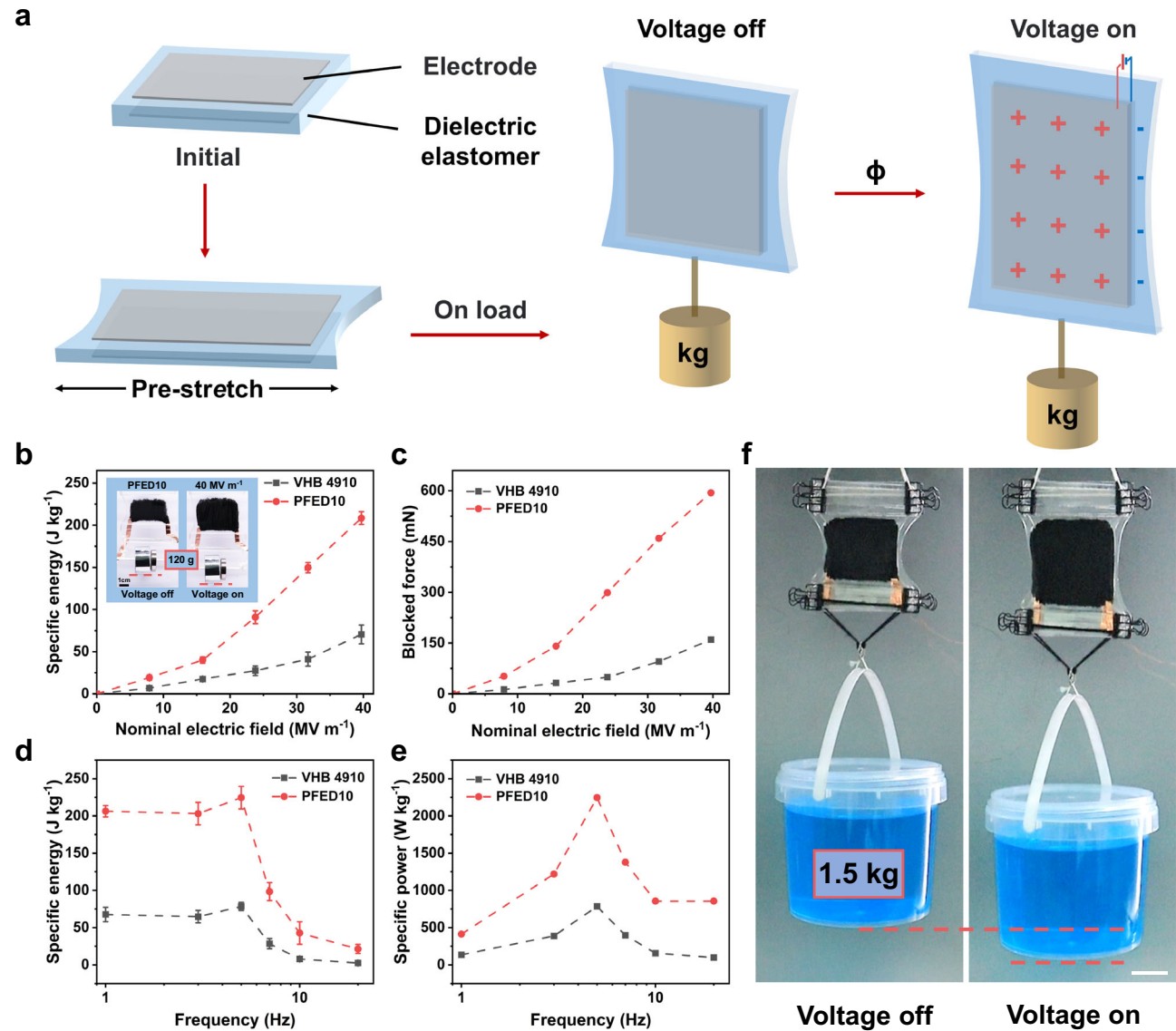

**Fig. 4 | Energy density and power density properties of PFED10. a** Schematic illustration of a pure-shear DEA as a linear actuator to test the specific energy and specific power. The DEA was pre-stretched laterally and loaded in the perpendicular direction. **b** Specific energy of VHB 4910 and PFED10 films at different electric fields with a 120 g load. Inset: photographs of linear actuation of PFED10 pure-shear DEA at 40 MV m⁻¹ with a 120 g load. The frequency was fixed at 0.5 Hz. Scale bar, 1 cm. Error bars show s.d., $n = 3$. **c** Block force of VHB 4910 and PFED10 films at different electric fields with a 120 g load. **d** Specific energy of VHB 4910 and PFED10 films at different frequencies. The driving electric field was fixed at 40 MV m⁻¹. Error bars show s.d., $n = 3$. **e** Average specific power density of VHB 4910 and PFED10 films at different frequencies. **f** Photographs of two scaled-up pure-shear DEAs prepared by PFED10 films can output large forces to lift a 1.5 kg bucket. Scale bar, 3 cm.

acrylate (DA) were purchased from Energy Chemical. 2-Hydroxy-2-methylpropiophenone (HMPP) as the radical photo-initiator was purchased from Energy Chemical. All the commercially available reagents were used without further purification. Carbon grease (NyoGel 756 G, Nye Lubricants) was used as compliant electrodes. VHB 4910 (3 M company) was used as DE for contrast.

## Preparation of DE films
For PFED10 films, 15 mmol HFBA, 1.5 mmol EA, 1.5 mmol DA, and 0.09 mmol HMPP were mixed in a brown glass bottle and stirred overnight to obtain a well-dispersed precursor solution. Then the precursor solution was poured into silicone molds. After UV cured in $N_2$ for 1 h (UV light: 320-400 nm, 36 W), transparent DE films were obtained. For other PFED series, PF, PE, PD, and PFE films, the molar ratios of monomers and photo-initiator were shown in Supplementary Table 1 and the preparation process was similar to that of PFED10.

## Mechanical properties characterization
Tensile tests were conducted using Sunstest UTM2502 at room temperature with dumbbell-shaped samples of 2 mm wide, 1 mm thick, and 12 mm long at a loading rate of 100 mm min⁻¹. The cyclic tensile stress-strain curves were obtained at a loading rate of 100 mm min⁻¹ and the samples were subjected to various strains from 10% to 300% with the interval of 1 min. Young's modulus was calculated from the initial slope of the stress-strain curves at 5% strain. Self-healing tests were performed at room temperature. To clearly observe the self-healing process, the samples were stained with rhodamine B. The samples were cut using a sharp knife underwater and then the cut pieces were gently re-contacted. The whole process took place in the air or underwater and left material healing in the air or underwater for a certain time. Mechanical tests were conducted after the designated healing time at a loading rate of 100 mm min⁻¹. Each mechanical test was repeated with at least three individual samples.

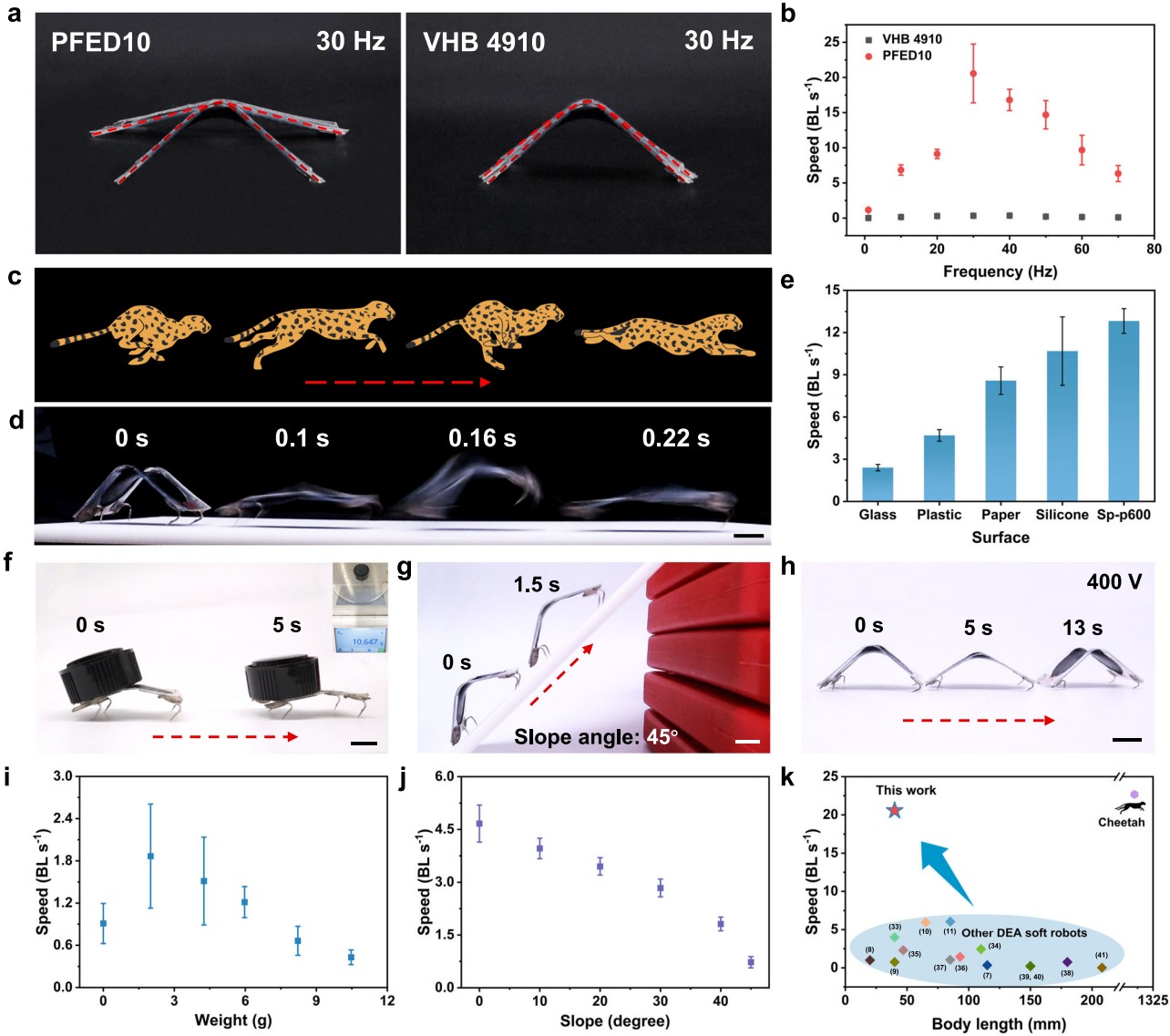

**Fig. 5 | Fast running robot fabricated by PFED10. a** Snapshots of the PFED10-based and VHB 4910-based soft robots vibrations at 38 MV m$^{-1}$ and 30 Hz. **b** The running speeds of soft robots as a function of frequency at 38 MV m$^{-1}$. Error bars show s.d., $n = 3$. **c** Illustration depicting the running posture of a cheetah. **d** Snapshots of the PFED10-based soft robot running with ultrafast speed at 38 MV m$^{-1}$ and 30 Hz. Scale bar, 1 cm. **e** The running speeds of PFED10-based soft robot on different substrates. Error bars show s.d., $n = 3$. **f** The PFED10-based soft robot carried a load of 10.6 g, which was about 17 times its own body weight at a speed of 0.43 BL s$^{-1}$. Scale bar, 1 cm. **g** The PFED10-based soft robot climbed a slope of 45° with a speed of 0.73 BL s$^{-1}$. Scale bar, 1 cm. **h** The PFED10-based soft robot running at a low voltage of 400 V. Scale bar, 1 cm. **i** The running speeds of PFED10-based soft robot as a function of load weight. Error bars show s.d., $n = 3$. **j** The running speeds of PFED10-based soft robot as a function of slope angle. Error bars show s.d., $n = 3$. **k** The maximum running speeds of DEA-based soft robots and cheetah versus body length.

## Dielectric properties characterization

Dielectric properties as a function of frequency were measured by a Broadband Dielectric Spectroscopy (Alpha-T, Novocontrol Technologies GmbH & Co. KG) at room temperature. The samples were cut into circles with a diameter of 10 mm and a thickness of 1 mm, and then placed between parallel electrodes. Frequency sweep range was from 10 Hz to $10^6$ Hz. The conductivity as a function of temperature was similarly tested from −20 °C to 100 °C. The electrical breakdown strength of dielectric elastomer was measured by a high voltage tester (BDJC-50kV, Beijing, Beiguang). The dielectric elastomers were placed between two 10-mm-diameter parallel electrodes, which were soaked in the silicone oil at room temperature, and then a DC voltage ramp of 500 V s$^{-1}$ was applied to the electrodes until breakdown. We tested ten samples for initial and healed PFED10 without pre-stretch and further fitted the breakdown strength data to the Weibull distribution

function, $F(E_B) = 1\text{-}\exp(\text{-}(E_B/\eta)\ \beta)$, where $F(E_B)$ is the cumulative probability of electric failure, $E_B$ is the electrical breakdown strength measured for each sample in the experiments, $\eta$ is the characteristic breakdown strength which is determined from the distribution at which 63.2% of the films have broken down electrically, and $\beta$ is the shape parameter that evaluates the scatter of data.

## Other characterizations

Transmission data was obtained using a HITACHI UH4150 spectrophotometer. $^1$H NMR spectrum was measured on a JEOL JNM-ECZ400S spectrometer (400 MHz) at room temperature. Solid-state $^{13}$C NMR and $^{19}$F NMR were measured on a JEOL JNM-ECZ600R spectrometer (600 MHz). Fourier transform infrared (FTIR) spectra were carried out using Nicolet 6700FTIR with transmission mode. Raman spectra were performed using Horiba HR Evolution with

a 532 nm laser (50 mW) through an objective lens. Gel permeation chromatography (GPC) measurement was performed by Waters 515 (Milford, MA) using polystyrene (PS) as standard and dimethyl formamide (DMF) as eluent. Water contact angles on the surfaces of the materials were measured by a drop shape analyzer (Dataphysics OCA15Pro). Differential scanning calorimetry (DSC) measurements were performed using TA Instruments DSC 250 from −80 °C to 150 °C with a heating rate of 10 °C min⁻¹. Dynamic mechanical analysis (DMA) measurements were performed using TA Instruments DMA 850 from −60 °C to 60 °C with a temperature ramping rate of 2 °C min⁻¹ at a frequency of 1 Hz and a strain of 1%. Rheology measurements were conducted using TA Instruments ARG2 with a parallel plate of 8 mm diameter. The thickness of samples was fixed to 1 mm. Oscillatory frequency-sweep measurements were performed at a strain of 1% with the frequency ranging from 0.1 to 400 rad s⁻¹. Wide-angle X-ray diffraction (WAXD) and small-angle X-ray diffraction (SAXD) were performed using D/max-2550. All the photos were taken by Canon M50.

## Fabrication of DEAs
For the DEA without pre-stretch, a 0.5 mm thick DE film was fixed on an annular PMMA rigid frame with an inner diameter of 20 mm. Carbon grease as the compliant electrodes with a 20-mm diameter was coated on both sides of the film. For the DEA with pre-stretch, a 2 mm initial thick DE film was pre-stretched radially to 2.75 times the original radius, and fixed between two annular rigid PMMA frames with an inner diameter of 70 mm and outer diameter of 110 mm to maintain its pre-stretch. Carbon grease as the compliant electrodes with a 10-mm diameter was coated on both sides of the film. For low-voltage DEAs, the DE films with an initial thickness of 0.25 mm or 0.45 mm were prepared and also pre-stretched radially to 2.75 times the original radius. The DEAs were powered by high-voltage source, which was generated by a signal generator (UTG1005A) and a high-voltage amplifier (Model 615-10, TREK).

## Fabrication of soft robots
Firstly, a 2 mm initial thick DE film was pre-stretched radially to 2.75 times the original radius, and fixed between two annular rigid PMMA frames with an inner diameter of 70 mm and outer diameter of 110 mm to maintain its pre-stretch. The stiff PET frames with thicknesses of 0.1 mm and 0.2 mm were cut into specific shapes using the laser cutting system (Epilog Legend 36 EXT, Chinese Laser Systems). Then, a 0.1 mm thick PET frame was attached to the one side of the DE film and a 0.2 mm thick PET frame as a constitute that restricted the direction of muscle locomotion was stacked on the other side of the film. Carbon grease was coated on both sides of the film as compliant electrodes. After removing the rigid PMMA frames and surrounding DE film, the pre-stretched muscle was contracted and constrained by the 0.2 mm thick PET frame, and the body assembly was bent into a buckled shape with a characteristic curvature. Finally, four stiff metallic hooks were sticked on the soft robot's body as front and hind feet, pointing in the backward direction of the robot. For the low-voltage robot, the fabrication process was similar except for the initial thickness of DE film (0.45 mm) and the thickness of PET frames (0.075 mm and 0.1 mm). The soft robots were also powered by the high-voltage source.

## Static actuation performance measurement
For the DEA without pre-stretch, the DE film produced a dome-like area expansion in the out-of-plane direction when the high voltage was applied. The actuation was recorded by camera (Canon M50). According to the camera images, the area strains ($S$) were calculated by the following equation $S = (h/r)^2 \times 100\%$, where $h$ is the height of the dome and $r$ is the radius of compliant electrodes. The nominal electric field was calculated by dividing the applied voltage by the initial

thickness of the DE film before actuated. The nominal electric field applied on electrodes was increased from 0 to 15 MV m⁻¹ at 0.1 Hz.

For the DEA with pre-stretch, the DE film expanded in area and decreased in thickness when the high voltage was applied. The actuation was recorded by camera (Canon M50). According to the camera images, the area strains were analyzed by the software Image J and calculated through the following equation $\frac{a_1 - a_0}{a_0} \times 100\%$, where $a_1$ is the actuated area under high voltage, and $a_0$ is the initial area. The nominal electric field applied on the electrodes was increased from 0 to 50 MV m⁻¹ and each nominal electric field was held for 30 s. The low-voltage DEAs were driven from 0 to 600 V at 0.2 Hz.

## Dynamic actuation performance measurement
Dynamic actuation performance measurements were performed using the DEAs with pre-stretch. For frequency response, the nominal electric field applied on the DEAs was fixed at 37.5 MV m⁻¹ and the frequency varied from 1 Hz to 100 Hz. For time response, the DEAs were driven by a nominal electric field of 25 MV m⁻¹ for 500 s. For the cycle tests, the nominal electric fields applied on the DEAs were 10.8 MV m⁻¹ or 37.5 MV m⁻¹ and the frequency was fixed at 1 Hz.

## Self-healing actuation performance measurement
The DE film was cut using a sharp knife and then the cut pieces were gently re-contacted underwater and healed in water for a certain time. For the DEA without pre-stretch, the healed DE film was fixed on an annular PMMA rigid frame with an inner diameter of 20 mm. Carbon grease as the compliant electrodes with a 20-mm diameter was coated on both sides of the film. For the DEA with pre-stretch, the healed DE film was pre-stretched radially to 1.25 times the original radius, and fixed between two annular rigid PMMA frames with inner diameter of 35 mm and outer diameter of 50 mm to maintain its pre-stretch. Carbon grease as the compliant electrodes with a 10-mm diameter was coated on both sides of the film. The nominal electric field applied on the electrodes was increased from 0 to 20 MV m⁻¹ at 0.1 Hz. The actuation strain was recorded by camera (Canon M50).

## Specific energy and specific power measurements of DE films in pure-shear mode
The DE film with 2 cm width, 3 cm length, and 0.5 mm initial thickness was pre-stretched laterally to 3 times the original width and fixed by rigid PMMA frames (6 cm width) to maintain lateral pre-stretch. Then, the DE film loaded 60 g or 120 g to measure specific energy and specific power. For loading 60 g, the active area of DE film is about 4 cm long and 1.2 cm wide. For loading 120 g, the active area of DE film is about 4 cm long and 2.5 cm wide. The specific energy at different nominal electric fields was tested by applying increasing nominal electric fields from 0 to 40 MV m⁻¹ at 0.5 Hz. The specific energy and specific power at different frequencies were tested by applying increasing frequencies from 1 to 100 Hz at 20 MV m⁻¹ or 40 MV m⁻¹. The specific energy ($E$) and specific power ($P$) of DE films were calculated according to the increase of load potential energy during the film contraction, as shown in the following equation[6,26] $E = \frac{m_L g h}{m_a}$, $P = 2E \times f$, where $m_L$ is the mass of load, $m_a$ is the mass of active area, $h$ is the increased height of the load, $f$ is the frequency. Cycle test for specific energy and specific power was performed with a load of 100 g at 28 MV m⁻¹ and 5 Hz. In order to lift the 1.5 kg bucket, we prepared two dielectric elastomer films with 3 cm width, 4 cm length, and 3.5 mm initial thickness. Lateral pre-stretch was also fixed at 3 times. The active area of DE film is about 6 cm long and 5.3 cm wide. The blocked force was measured by constraining the pure-shear DEAs on a universal tensile testing machine (Sunstest UTM2502) from 0 to 40 MV m⁻¹ at 0.5 Hz. When the DEAs were driven, their deformation was fully constrained and a maximum force was generated, which was recorded as blocking force.

## Performance characterization of soft robots

The soft robots were driven at 38 MV m$^{-1}$ and 30 Hz for fast moving. When carrying load, the soft robots were driven at 35 MV m$^{-1}$ and 10 Hz. The soft robots climbed a slope at 35 MV m$^{-1}$ and 10 Hz. The locomotion of low-voltage robot was driven at 400 V and 10 Hz. The movies of soft robots moving were all recorded by Canon M50.

## Data availability

The data that support the findings of this study are available within this article and its Supplementary Information. All data are available from the corresponding author upon request.

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

## Acknowledgements

This work was supported by the National Natural Science Foundation of China (Grant No. 21890731 (C.W.) and Grant No. 22075164 (C.W.)).

## Author contributions

C.W. and W.F. conceived the concept and designed the experiments. W.F. carried out experiments, collected the data, and wrote the paper. L.S., Z.J., L.C., Y.L. and H.X. provided advice to the manuscript. All authors reviewed and revised the manuscript.

## Competing interests

The authors declare no competing interests.
