## [Peer Review File · Nature Communications]

REVIEWER COMMENTS

Reviewer #1 (Remarks to the Author):

The authors report synthesizing a novel dielectric elastomer with a dielectric permittivity of 10 and a large actuation strain at a low electric field.

The synthesis starts from 2,2,3,4,4-hexafluorobutyl acrylate (HFBA), 2-ethylhexyl acrylate (EA), and dodecyl acrylate (DA) and uses a one-step UV-induced photopolymerization. The mechanical properties of the resulting polymers were tuned by using different amounts of monomers. One material was selected for further characterizations, PFED10, and was used for constructing actuators and a small robot. While the material presented exhibited good performance, the referee cannot recommend publishing this work in its current form.

The authors mentioned making a random copolymer but failed to provide molecular structure and composition information. Given this, how can the author explain the presence of nanodomains by SAXD? The authors should provide sufficient information to demonstrate they have a random copolymer. In addition, the ^1H and ^{13}C NMR of the polymers should be provided, as well as the molecular weight and distribution. No purification of the polymers was done after the synthesis. Therefore, the authors should provide evidence that the polymerization is conducted to full conversion.

The dielectric permittivity of the material is given; however, the values of the conductivity at different frequencies should also be included. Additionally, the authors should provide temperature-dependent impedance spectroscopy measurements to assess better the temperature window where this material can be used.

DSC measurements from very low temperatures of -80 up to 150 °C should be provided to verify the presence of any phase transition.

The mechanism behind healing underwater should be properly explained.

It is mentioned that the PFED10 elastomer possessed a Young's modulus of 0.09 MPa. At which strain was this measured?

Figure S5 shows the dielectric losses at different frequencies. The authors should mention how the losses are influencing the actuation.

Figure S6 shows the strain curve of PD and not of PFED10. Please explain.

The mechanical losses $\tan \delta = 0.41$ at 1 Hz of PFED10 is relatively high. How does this affect the actuation at different frequencies?

According to Figure S9, material PFED10 exhibits a storage modulus of about 10^5 MPa. Please explain.

The authors mentioned that the electromechanical sensitivity (β), defined as the ratio of dielectric constant to Young's modulus, is an important parameter to evaluate the theoretical actuation

performance of DEs. As shown in Fig. 3a, the electromechanical sensitivity of PFED10 is up to 114, more than 7 times higher than that of VHB 4910 ($\beta = 15$).

A comparison should be given with the state-of-the-art high dielectric permittivity elastomers, including electromechanical sensitivity. See Niu et al., *Journal of Polymer Science Part B: Polymer Physics*, 2013, 51 (3), 197-206; Caspari et al. *Journal of Materials Chemistry C*, 2018, 6 (8), 2043-2053; Sheima et al. *Macromolecular rapid communications* 2019, 40 (16), 1900205; Bele et al. *Journal of Applied Polymer Science* 139 (22), 52261, 2022.

Suppose the sensitivity of the developed elastomer is 7 times higher than VHB. In that case, the authors should explain why this is not reflected in the same performance increase, given that the losses in PFED10 are smaller than in VHB 4910. Additionally, the actuation is inferior to some reported materials.

Which is the maximum achieved actuation at the dielectric breakdown when the actuator is not prestrained?

Why did the authors choose to conduct a 2.75×2.75 biaxial pre-stretch?

A maximum area strain of 253% at $46 \text{ V } \mu\text{m}^{-1}$ was achieved for the new material, which was almost 14 times larger than VHB 4910. Given that the sensitivity factor was only 7 times higher, an explanation is needed for this improvement.

Figure 3e. Why is there such a huge difference in the actuation of films with different thicknesses?

Cyclic test for the unrestrained membrane should also be given.

Correct $\text{V } \mu\text{m}^{-1}$ to standard units.

Finally, the English language could be refined for greater clarity and coherence.

Reviewer #2 (Remarks to the Author):

In "A large-strain and ultrahigh energy density dielectric elastomer for fast moving soft robot", Feng et al. report a new elastomeric material that enables high specific energies at relatively low electric fields. The authors extensively characterize the mechanical and the actuation properties of the material. The presented material far exceeds previously reported values of specific energy. These extraordinary properties enable lifting large loads and fast robots.

The presented work provides a clear advance for dielectric elastomers, so it will be highly interesting to the readership of Nature Communications. The work is well described and the authors provide all important metrics that are relevant for the dielectric elastomer community. I thus have only few comments, which I recommend the authors to address, before accepting the article:

1. It is very interesting that the specific energy reported here for VHB, which has been a standard material for DEAs for a long time, is comparable to the specific energy of PHDE (ref. 6), a material that was recently reported in Science to provide a large increase in specific energy compared to other

materials. I am wondering how much the experimental conditions influence the measured performance metrics and how comparable values from different article really are. Please discuss.

2. The authors report that their material is self-healing. How much does the actuation performance change after self-healing (e.g., maximum actuation strain, breakdown field)?

Minor

1. Throughout the article, the authors report performance metrics normalized by the mass of the material and term it "... density" (e.g., energy density). This term is normally reserved for metrics that are normalized by their volume. For normalization by mass, the term "specific ..." is used (e.g., specific energy). Please change the terms throughout the text.

2. I recommend to also show the measured performance metrics for VHB in Figure 1c,d as they are also very high compared to most previous literature.

3. Some references to supplementary figures seem to be incorrect (e.g., page 7, line 137 refers to Supplementary Figs. 12 and 13, but should refer to Supplementary Figs. 14 and 15).

Reviewer #3 (Remarks to the Author):

To achieve high dielectric constant and low modulus of DEs, the authors introduce polar fluorinated groups and nanodomains aggregated by long alkyl side chains into DE design, which simultaneously achieves large actuation strain and high energy density under a low driving electric field. The new DE exhibits a maximum area strain of 253% at a low driving electric field of 46 V μm^{-1} . Notably, it achieves an ultrahigh energy density of 225 J kg^{-1} at only 40 V μm^{-1} , around 66 times that of natural muscle and twice that of the state-of-the-art DE, and the corresponding power density reaches 2245 W kg^{-1} . Moreover, the DE has excellent cycle stability and good performance for soft robot application. The work reaches a high level, and the designed DE is promising for electric field actuation such as soft robot application. There are some questions that need to be addressed before publication.

1. The language should be double-checked. Some obvious mistakes are as follows:

1) Line 14, "endow DE with high dielectric constant and desirable modulus"

2) Line 98, "elastomer is expected to displays large actuation strain"

3) Lines 94-95, "To tune the mechanical and dielectric properties of our DE, the molar ratio of HFBA, EA 94 and DA were systematically changed"

4) Line 130, "good elasticity of PFED10 were arise from the nanodomains"

5) Lines 131-133, "Compared to PFE, the SAXD result of PFED10 131 were featured with a peak like PD", "which indicated the consist of nanodomains and the 132 average distance d ($d = 2\pi/q_{\max}$) between the nanodomains was about 3.76 nm"

6) Line 178, "In addition, the PFED10-based DEA have self-healing property"

7) ...

2. Confirm if the expression is correct "The stress-strain curve of PE is soft and weak or brittle" in Supplementary Fig. 2.

3. According to the performance of the samples, the fraction of DA is critical to the properties. Why is molar ratio of 10 the best? More discussion should be given.

4. For all PFED series, DE content is constant, why?

Dear Reviewers,

Thank you very much for your time and effort you have put in reviewing our manuscript “**A large-strain and ultrahigh energy density dielectric elastomer for fast moving soft robot**”. We are grateful for all the valuable comments and suggestions that you offered to us to improve the quality of the manuscript. We have responded to all the questions point to point in the following section. For your convenience, the original comments are reproduced in **blue**, our response is in **black**, and changes made to the manuscript are highlighted in **red**. As you will find, our manuscript has been updated in the following aspects:

1) More experiments have been added to characterize the composition, molecular weight, and conversion of our dielectric elastomer including the ^1H NMR, solid-state ^{13}C NMR, solid-state ^{19}F NMR, FTIR spectra, Raman spectra, DSC, and GPC.

2) Additional experimental results have been added to show the conductivity of our dielectric elastomer varies with frequency as well as temperature.

3) The cyclic actuation test for unrestrained film has been performed to show the cycle stability of our dielectric elastomer. The supplementary table for the electromechanical sensitivity of this work and other dielectric elastomers has been added to the revised manuscript.

4) The electrical breakdown strength of our dielectric elastomer before and after self-healing without pre-stretch has been measured to show the actuation performance change after self-healing.

5) More experimental details and a few key references have been updated to the revised manuscript.

6) We have corrected all the typos and language expression errors pointed out by the reviewers and made revisions after carefully re-reading the manuscript.

We wish our responses can answer your questions and alleviate your concerns. More specific point-to-point responses are as follows.

Response to Reviewer #1

Overview. *The authors report synthesizing a novel dielectric elastomer with a dielectric permittivity of 10 and a large actuation strain at a low electric field.*

The synthesis starts from 2,2,3,4,4,4-hexafluorobutyl acrylate (HFBA), 2-ethylhexyl acrylate (EA), and dodecyl acrylate (DA) and uses a one-step UV-induced photopolymerization. The mechanical properties of the resulting polymers were tuned by using different amounts of monomers. One material was selected for further characterizations, PFED10, and was used for constructing actuators and a small robot. While the material presented exhibited good performance, the referee cannot recommend publishing this work in its current form.

Response: We thank the reviewer for the positive comments.

Comment 1. The authors mentioned making a random copolymer but failed to provide molecular structure and composition information. Given this, how can the author explain the presence of nanodomains by SAXD? The authors should provide sufficient information to demonstrate they have a random copolymer. In addition, the ^1H and ^{13}C NMR of the polymers should be provided, as well as the molecular weight and distribution. No purification of the polymers was done after the synthesis. Therefore, the authors should provide evidence that the polymerization is conducted to full conversion.

Response 1: The PFED10 copolymer is prepared by conventional photo-initiated radical polymerization with a mixture of acrylate-based monomers, which has been used to obtain random copolymers in many previous works (*Adv. Mater.* 2021, 33, 2006111; *Adv. Mater.* 2021, 2105306; *Nat. Mater.* 2022, 21, 359-365; *Nat. Commun.* 2023, 14, 4853). Random copolymers can form nanodomains through the aggregation of molecular chains (*ACS Appl. Polym. Mater.* 2019, 1, 359-368; *Nat. Commun.* 2022, 13, 2279; *Adv. Funct. Mater.* 2022, 32, 2107538; *Adv. Funct. Mater.* 2022, 32, 2112293). For example, in one previous work by Li et al., the polymethyl methacrylate segments in random copolymer can aggregate into loose nanodomains and serve as stiff but dynamic crosslinkers (*Nat. Commun.* 2022, 13, 2279). In our copolymer, the long alkyl side chains in DA can aggregate to form nanodomains and serve as dynamic physical crosslinkers, which was supported by the stress-strain curves and SAXD result of PFED10 (Figs. 2a and 2g).

To further confirm and characterize the structure and composition of the PFED10 copolymer, we have performed additional experiments including Nuclear magnetic resonance (NMR), Gel permeation chromatography (GPC), Fourier transform infrared (FTIR) spectroscopy, Raman spectroscopy, Differential scanning calorimetry (DSC), and Thermogravimetric analysis (TGA). As shown in Figs. R1, R2, and R3, the ^1H , solid-state ^{13}C , and solid-state ^{19}F NMR spectra of the PFED10 copolymer contained all the structural features of the three monomers, indicating that the copolymer was composed of HFBA, EA, and DA. In the ^1H NMR and solid-state ^{13}C NMR spectra of the PFED10 copolymer, the typical chemical shifts of $\text{CH}_2=\text{CH}-$ (from the comonomers) all disappeared, demonstrating that the comonomers were almost completely converted and the copolymer was successfully obtained. According to the results of the ^1H and solid-state ^{13}C NMR spectra of PFED10, except for the conversion of vinyl, no other special chemical shifts were observed and the DSC curve of PFED10 showed only one glass transition temperature ($T_g = -22.1\text{ }^\circ\text{C}$), all of which indicated that PFED10 was a random copolymer (Figs. R1, R2, and R9). Furthermore, the molecular weight of the PFED10 copolymer was confirmed by GPC. As shown in Fig. R4, the M_w and PDI of the PFED10 copolymer were 37761 Da and 1.21, respectively.

To verify the full conversion of the polymerization, we added FTIR spectroscopy, Raman spectroscopy, and TGA tests. As shown in Fig. R5a, the FTIR spectra of the precursor of PFED10 before and after polymerization showed that the absorption peak corresponding to the $\nu(\text{C}=\text{C})$ (from acrylate) at 1637 cm^{-1} vanished, indicating that the conversion of the comonomers was almost 100%. In Raman spectra, the Raman band of $\text{C}=\text{C}$ bond at 1643 cm^{-1} in monomers vanished in the corresponding copolymer indicating the full conversion of the comonomers (Fig. R5b). In addition, the temperature at 5% of weight loss (T_{d5}) for PFED10 was $324\text{ }^\circ\text{C}$. The boiling points of three comonomers were all below $250\text{ }^\circ\text{C}$ and the mass loss of the polymer before $250\text{ }^\circ\text{C}$ was only 0.5% which indicated a high solid content of the PFED10 copolymer (Fig. R6). The results of NMR spectra, FTIR spectra, Raman spectra, and TGA were consistent, which proved that the monomers

were fully converted.

We have added Figs. R1-R5 to the Supporting Information as **Supplementary Figs. 7-11** and made the corresponding description in the manuscript.

Fig. R1 ¹H NMR (DMF-d₇) spectrum of PFED10.

Fig. R2 Solid-state ¹³C NMR spectrum of PFED10.

Fig. R3 Solid-state ¹⁹F NMR spectrum of PFED10.

Fig. R4 GPC curve of PFED10.

Fig. R5 FTIR and Raman spectra of PFED10 and PFED10 precursor. The PFED10 precursor was a mixture of HFBA, EA, and DA with a molar ratio of 100: 10: 10.

Fig. R6 TGA curve of PFED10. The temperature at 5% of weight loss (T_{d5}) for PFED10 was $324 \text{ }^\circ\text{C}$.

[Revised manuscript]

“Therefore, considering the mechanical properties, dielectric properties, and actuation area strain, the PFED10 was selected for our study. The results of ^1H NMR, solid-state ^{13}C NMR, and solid-state ^{19}F NMR characterized the chemical structure of the PFED10 (Supplementary Figs. 7, 8, and 9). In Fourier transform infrared (FTIR) and Raman spectra, the disappearance of $\nu(\text{C}=\text{C})$ (from comonomers) at 1637 cm^{-1} and 1643 cm^{-1} , respectively, indicated that the conversion of comonomers was almost 100 % (Supplementary Fig. 10). In addition, the weight-average molecular weight (M_w) of PFED10 was 37761 Da (Supplementary Fig. 11). The PFED10 elastomer possessed Young’s modulus of 0.09 MPa (Fig. 2c), around one-third of that of commercial VHB 4910 (~ 0.32 MPa).”

“Other Characterizations: Transmission data was obtained using a HITACHI UH4150 spectrophotometer. ^1H NMR spectrum was measured on a JEOL JNM-ECZ400S spectrometer (400 MHz) at room temperature. Solid-state ^{13}C NMR and ^{19}F NMR were measured on a JEOL JNM-ECZ600R spectrometer (600 MHz). Fourier transform infrared (FTIR) spectra were carried out using Nicolet 6700FTIR with transmission mode. Raman spectra were performed using Horiba HR Evolution with a 532 nm laser (50 mW) through an objective lens. Gel permeation chromatography (GPC) measurement was performed by Waters 515 (Milford, MA) using polystyrene (PS) as standard and dimethyl formamide (DMF) as eluent. Water contact angles on the surfaces of the materials were measured by a drop shape analyzer (Dataphysics OCA15Pro).”

Comment 2. The dielectric permittivity of the material is given; however, the values of the conductivity at different frequencies should also be included. Additionally, the authors should provide temperature-dependent impedance spectroscopy measurements to assess better the temperature window where this material can be used.

Response 2: As suggested, the values of the conductivity at different frequencies have been added as shown in Fig. R7. In the range of 10 Hz to 10^6 Hz, the conductivity of PFED10 is similar to that of VHB 4910. PFED10 was considered to have a low conductivity and was shown to be electrically insulating. In addition, we have tested the conductivity of PFED10 at different temperatures. As shown in Fig. R8, the conductivity of PFED10 changed slightly and PFED10 remained electrically insulating from $-20\text{ }^\circ\text{C}$ to $100\text{ }^\circ\text{C}$ indicating a wide operating temperature range.

We have added Figs. R7 and R8 to the Supporting Information as **Supplementary Figs. 13 and 14**, respectively, and made the corresponding description in the manuscript.

Fig. R7 Conductivity of VHB 4910 and PFED10 in the range of 10 Hz to 10⁶ Hz.

Fig. R8 Conductivity of PFED10 from -20 °C to 100 °C.

[Revised manuscript]

“Owing to the highly polar CF₃ groups in HFBA segments, PFED10 elastomer exhibited a high dielectric constant of 10.23 at 1 kHz, which was much higher than that of VHB 4910 (4.77, at 1 kHz) (Fig. 2b). With the increase of frequency, the dielectric losses of PFED10 and VHB 4910 generally increased, which led to the decrease of the total electro-mechanical efficiency and actuation area strain (Supplementary Fig. 12). However, compared with dielectric losses, mechanical losses dominate the total electro-mechanical efficiency⁴⁶. In addition, we tested the conductivity of PFED10. PFED10 was shown to be electrically insulating in the range of 10 Hz to 10⁶ Hz and the conductivity of PFED10 changed slightly from -20 °C to 100 °C indicating a wide operating temperature range (Supplementary Figs. 13 and 14).”

Comment 3. DSC measurements from very low temperatures of -80 up to 150 °C should be provided to verify the presence of any phase transition.

Response 3: We have added the DSC measurement for PFED10 from -80 to 150 °C. As shown in Fig. R9, the DSC curve of PFED10 exhibited a glass transition temperature (T_g) of -22.1 °C without obvious phase transition. Aggregate structure is weaker than crystallization making it difficult to observe obvious heat changes on the DSC curve. According to the result of SAXD, our small-size nanodomains aggregated by the long alkyl side chains in DA segments existed and the average distance between the nanodomains was only 3.76 nm (Fig. 2g). The mechanical test results can also support the existence of nanodomains. With the increase of DA, the copolymers exhibited obvious strain-hardening behavior and the elongation at break decreased (Fig. 2a). Therefore, a single T_g on the DSC curve could not mean that the copolymers have no nanodomain aggregate structure. The similar phenomenon can be seen in previous reports as well (*ACS Appl. Polym. Mater.* 2019, 1, 359-368; *Nat. Commun.* 2022, 13, 2279). We have added Fig. R9 to the Supporting Information as **Supplementary Fig. 22**.

Fig. R9 DSC curve of PFED10.

Comment 4. The mechanism behind healing underwater should be properly explained.

Response 4: We summarize the mechanism of PFED10 underwater self-healing into the following three important factors:

i) Strong dipole-dipole interactions as dynamic bonds. Due to the highest electronegativity of fluorine, the C-F bond is highly polarized, which makes the C-F bond exhibit a significant ionic character rather than the typical electron sharing of a covalent bond. Therefore, the bonding constants for dipole-dipole interactions between the C-F bonds are very strong and they can interact with other groups through electrostatic (dipole-dipole and charge-dipole) interactions (*Chem. Soc. Rev.* 2008, 37, 308; *Nat. Commun.* 2022, 13, 1338). In our copolymer, the HFBA segments contain a high density of CF_3 dipoles. These highly polar parts can interact with each other to facilitate the self-healing process (Fig. R10).

ii) A sufficiently low glass transition temperature (T_g) makes the polymer chains flexible enough at room temperature to allow diffusion and re-entanglements of the polymer chains at

damaging locations. Based on the DSC result, our PFED10 had a T_g of $-22.1\text{ }^\circ\text{C}$ well below room temperature (Fig. R9). As a result, the local diffusion of polymer chains across damaged interfaces could easily occur at room temperature. Meanwhile, the abundant highly polar groups on the freshly fractured interface could form interactions to facilitate the diffusion of polymer chains, which promoted the healing process of the material (Fig. R10).

iii) Fluorinated polymers are hydrophobic and chemically stable in aqueous environments because the C-F bond is a very poor hydrogen bond acceptor (*Adv. Mater.* 2018, 30, 1804602; *Nat. Electron.* 2019, 2, 75-82). The hydrophobic surfaces formed by the abundant F element facilitated the formation of a water-resistant molecular bridge between the fractured parts, and the nature of self-hydrophobization would allow the growth of hydrophobic interactions to further repel water molecules away from the interface (*Proc. Natl. Acad. Sci. USA.* 2013, 110, 15680), endowing the material with outstanding underwater self-healing behavior. As shown in Supplementary Fig. 20, PFED10 exhibited a water contact angle of 122° , indicating excellent hydrophobicity. Therefore, the influence of the water environment on the bonding constant of dipole-dipole interactions between C-F bonds was very limited, and PFED10 can self-heal underwater.

In summary, the combination of fast segment diffusion, abundant dipole-dipole interactions formed by polar groups, and hydrophobicity on the fractured interfaces made the PFED10 self-heal underwater. We have revised the manuscript as follows.

Fig. R10 Illustration of the self-healing process of PFED10.

[Revised manuscript]

“In addition, the PFED10 elastomer possessed hydrophobicity and self-healing capability (Supplementary Figs. 20 and 21). The low glass transition temperature (Supplementary Fig. 22), high density of dipole-dipole interactions between CF₃ groups, and excellent hydrophobicity allowed the polymer chains of PFED10 to diffuse and re-entangle at the damaged locations without being affected by the aqueous environment⁴⁷, endowing PFED10 with underwater self-healing capability. Low Young’s modulus, high dielectric constant, and good elasticity show that the PFED10 elastomer can have excellent actuation performance.”

Comment 5. It is mentioned that the PFED10 elastomer possessed a Young's modulus of 0.09 MPa. At which strain was this measured?

Response 5: Young's modulus was calculated from the initial slope of the stress-strain curves at 5% strain. We have revised the manuscript as follows.

[Revised manuscript]

“Young's modulus was calculated from the initial slope of the stress-strain curves at 5% strain.”

Comment 6. Figure S5 shows the dielectric losses at different frequencies. The authors should mention how the losses are influencing the actuation.

Response 6: Dielectric losses will influence the electro-mechanical conversion efficiency and then affect the actuation area strain. However, in dielectric elastomers, the losses include mechanical losses and dielectric losses. The losses will reduce the electro-mechanical conversion efficiency, which leads to a decrease in actuation area strain. The total efficiency of a dielectric elastomer actuator depends on the mechanical efficiency (η_m) and electrical efficiency (η_e), which are defined as follows (*Proc. SPIE* 2000, 3987, 51-64):

$$\eta_m = \frac{1}{1 + \pi \tan \delta_m} \quad (1)$$

$$\eta_e = \frac{1}{1 + \pi \tan \delta_e} \quad (2)$$

where $\tan \delta_m$ and $\tan \delta_e$ are mechanical losses and dielectric losses, respectively. Since mechanical losses are usually greater than dielectric losses, the electrical efficiency is much higher than the mechanical efficiency, and mechanical losses dominate the total efficiency (*Mater. Lett.* 2006, 60, 3437-3440; *J. Macromol. Sci. Part B: Phys.* 2012, 51, 2093-2104). In our work, with the increase of frequency, dielectric losses and mechanical losses tended to increase, decreasing the efficiency and the actuation area strain (Fig. 2f, Supplementary Fig. 12, and Fig. R11). Although the dielectric loss of PFED10 is higher than that of VHB 4910, the total efficiency of VHB 4910 is lower than that of PFED10 because the mechanical loss of VHB 4910 increases more with frequency (Fig. 2f). Therefore, the higher total efficiency of PFED10 makes that the normalized actuation area strain of PFED10 is higher than that of VHB 4910 at different frequencies. Supplementary Fig. 5 in the initial manuscript has been renamed Supplementary Fig. 12 in the new manuscript and we have revised the manuscript as follows.

[Revised manuscript]

“Owing to the highly polar CF_3 groups in HFBA segments, PFED10 elastomer exhibited a high dielectric constant of 10.23 at 1 kHz, which was much higher than that of VHB 4910 (4.77, at 1 kHz) (Fig. 2b). With the increase of frequency, the dielectric losses of PFED10 and VHB 4910 generally increased, which led to the decrease of the total electro-mechanical efficiency and actuation area strain (Supplementary Fig. 12). However, compared with dielectric losses, mechanical losses dominate the total electro-mechanical efficiency⁴⁶. In addition, we tested the conductivity of

PFED10. PFED10 were shown to be electrically insulating in the range of 10 Hz to 10^6 Hz and the conductivity of PFED10 changed slightly from -20 °C to 100 °C indicating a wide operating temperature range (Supplementary Figs. 13 and 14).”

Comment 7. Figure S6 shows the strain curve of PD and not of PFED10. Please explain.

Response 7: Supplementary Fig. 6 in the initial manuscript has been renamed Supplementary Fig. 4 in the new manuscript. The cyclic stress-strain curve of PFED10 has already been shown in Fig. 2d. PFED10 always showed smaller hysteresis and residual strain than that of PFE and VHB 4910 (Fig. 2d). Compared with PFE, the addition of DA in PFED10 improves the elasticity of elastomer. To clarify the role of DA in improving the elasticity of PFED10, we measured the cyclic stress-strain curves of PF, PE, and PD (Supplementary Fig. 4). As shown in Supplementary Fig. 4, PD showed excellent elasticity compared to PF and PE, which indicated that DA contributed to the improving the elasticity of PFED10. To clarify, we have added the cyclic stress-strain curves of PFED10 and PFE to the updated **Supplementary Fig. 4** as follows.

Supplementary Fig. 4 Cyclic Stress-strain curves of PF, PE, PD, PFE, and PFED10. According to the cyclic stress-strain curves, PF and PE show poor elasticity. However, PD shows excellent elasticity, indicating that the aggregation of the long side chain serves as the physical crosslinkers. Compared with PFE, the addition of DA in PFED10 improves the elasticity of elastomer.

Comment 8. The mechanical losses $\tan \delta = 0.41$ at 1 Hz of PFED10 is relatively high. How does this affect the actuation at different frequencies?

Response 8: The mechanical loss characterizes the viscoelasticity of the material, which is critical to improving the frequency response of dielectric elastomers. At 1 Hz and 25 °C, the mechanical loss of PFED10 ($\tan \delta = 0.41$) was lower than that of VHB 4910 ($\tan \delta = 0.85$), suggesting that PFED10 had lower viscoelasticity than VHB 4910. With the increase of frequency, the mechanical losses of both PFED10 and VHB 4910 increased, but the mechanical losses of VHB 4910 were always much higher than that of PFED10 (Fig. 2f). Therefore, as shown in Fig. R11, the maximum actuation area strain of both PFED10 and VHB 4910 decreased with the increase of frequency due to the increase of mechanical losses. However, because the mechanical loss of PFED10 is smaller than that of VHB 4910, the decline of PFED10 is more gradual. The PFED10 film kept 66% of the

initial actuation area strain even at 100 Hz, showing a good frequency response. In sharp contrast, the normalized actuation area strain of VHB 4910 rapidly decayed below 50% at 30 Hz and maintained only 38% at 100 Hz. The low viscoelasticity allows the PFED10 to operate over a wide frequency range.

Fig. R11 Frequency response of VHB 4910 and PFED10 at 37.5 MV m^{-1} with 275% biaxial pre-stretch in the range of 1 Hz to 100 Hz. The actuation area strain at 1 Hz was normalized to 1.

Comment 9. According to Figure S9, material PFED10 exhibits a storage modulus of about 10^5 MPa . Please explain.

Response 9: We are sorry for this typo. Supplementary Fig. 9 in the initial manuscript has been renamed Supplementary Fig. 17 in the new manuscript. In Supplementary Fig. 17, the unit of storage modulus is Pa. We have revised the **Supplementary Fig. 17** as follows.

[Revised manuscript]

Supplementary Fig. 17. Storage modulus of VHB 4910 and PFED10 from $-55 \text{ }^\circ\text{C}$ to $55 \text{ }^\circ\text{C}$.

Comment 10. The authors mentioned that the electromechanical sensitivity (β), defined as the ratio of dielectric constant to Young's modulus, is an important parameter to evaluate the theoretical

actuation performance of DEs. As shown in Fig. 3a, the electromechanical sensitivity of PFED10 is up to 114, more than 7 times higher than that of VHB 4910 ($\beta = 15$).

A comparison should be given with the state-of-the-art high dielectric permittivity elastomers, including electromechanical sensitivity. See Niu et al., *Journal of Polymer Science Part B: Polymer Physics*, 2013, 51 (3), 197-206; Caspari et al. *Journal of Materials Chemistry C*, 2018, 6 (8), 2043-2053; Sheima et al. *Macromolecular rapid communications* 2019, 40 (16), 1900205; Bele et al. *Journal of Applied Polymer Science* 139 (22), 52261, 2022.

Response 10: Following the suggestion of the reviewer, we have compared the electromechanical sensitivity with the state-of-the-art high dielectric permittivity elastomers, as shown in Supplementary Table 2. PFED10 exhibited a high electromechanical sensitivity, exceeding that of most reported dielectric elastomers.

We have added **Supplementary Table 2** to the Supporting Information and made the corresponding description in the manuscript.

Elastomer	Dielectric constant	Young's modulus (MPa)	Electromechanical sensitivity (MPa ⁻¹)	References
E3-Cl-20	5.4	0.3	18	48
C0-P0	5.3	0.2	26.5	49
BmimSbF6	7.61	0.15	50.73	50
E-CL ₂	18	0.35	51.4	51
C2	10.1	0.154	65.6	52
Cl-8.5	5.2	0.07	74.3	53
BAC2	5.75	0.073	78.8	19
ECN	17.5	0.155	112.9	54
PFED10	10.23	0.09	114	This work

Supplementary Table 2. Electromechanical sensitivity of this work and other DEs.

[Revised manuscript]

“As shown in Fig. 3a, the electromechanical sensitivity of PFED10 is up to 114, which is more than 7 times higher than that of VHB 4910 ($\beta = 15$), exceeding that of most reported DEs⁴⁸⁻⁵⁴, suggesting excellent actuation performances of PFED10 (Supplementary Table 2).”

Comment 11. Suppose the sensitivity of the developed elastomer is 7 times higher than VHB. In that case, the authors should explain why this is not reflected in the same performance increase, given that the losses in PFED10 are smaller than in VHB 4910. Additionally, the actuation is inferior to some reported materials.

Response 11: The actuation principle of dielectric elastomers can be approximated as the lateral electrostatic compression and planar expansion of an incompressible linearly elastic material. According to the Maxwell stress, for small strains (<20%), the linear-elasticity and free boundary approximations were used to approximate the thickness strain (s_z) of dielectric elastomers (*Science* 2000, 287, 836-839):

$$s_z = -\frac{p}{Y} = -\frac{\varepsilon_r \varepsilon_0 E^2}{Y} = -\beta \varepsilon_0 E^2 \quad (1)$$

where ε_r is the relative dielectric constant, ε_0 is the dielectric constant of free space, E is the applied electric field, and Y is the elastic modulus. According to the above equation 1, for small strains, the electromechanical sensitivity (β) defined as the ratio of dielectric constant to Young's modulus is proportional to the thickness strain.

When the strains are greater than about 20%, Eq. 1 is unsatisfactory. For larger strains, while maintaining the assumption that the material is linearly elastic, Pelrine et al. showed that s_z can be approximated as follows (*Sens. Actuators A* 1998, 64, 77-85):

$$s_z = \frac{2}{3} + \frac{1}{3} \left[f(s_{z0}) + \frac{1}{f(s_{z0})} \right] \quad (2)$$

$$f(s_{z0}) = \left[2 + 27s_{z0} + \frac{[-4 + (2 + 27s_{z0})^2]^{\frac{1}{2}}}{2} \right]^{\frac{1}{3}} \quad (3)$$

$$s_{z0} = -\varepsilon_r \varepsilon_0 \frac{V^2}{Yz_0^2} \quad (4)$$

where V is the applied voltage, z_0 is the initial thickness of the dielectric elastomer film.

For a constant-volume material, the area strain (s_a) can be related to the thickness strain as follows:

$$s_a = \frac{1}{s_z + 1} - 1 \quad (5)$$

According to the above equations 2-4, for large strain, the electromechanical sensitivity is not proportional to the thickness strain. As a result, electromechanical sensitivity can only provide a theoretical foundation for evaluating DE behavior at small strains.

The definition of electromechanical sensitivity is based on the assumptions of small strains and linear elasticity, which limits its accuracy. Meanwhile, the actual actuation of dielectric elastomers is more complicated, such as the nonlinear of materials (for large strain), the viscoelasticity of materials, and experimental conditions (*Macromol. Rapid Commun.* 2009, 31, 10-36), which are not reflected in the electromechanical sensitivity. Therefore, the electromechanical sensitivity should be used to qualitatively reflect the actuation performances of dielectric elastomers. Deviations between actual actuation performance and electromechanical sensitivity are common (*J. Polym. Sci. B Polym. Phys.* 2013, 51, 197-206; *Adv. Funct. Mater.* 2015, 25, 2467-2475; *Polym. Chem.*, 2016, 7, 2709-2719; *J. Appl. Polym. Sci.* 2022, 139, 52261). For example, in one previous work by Tugui et al., their material X-5% can reach an electromechanical sensitivity of 23.5, 6 times that of Sylgard 186 (4), while the thickness strain of X-5% is 1.5 times that of Sylgard 186 (*Polym. Chem.*, 2016, 7, 2709-2719).

In our work, because PFED10 has higher electromechanical sensitivity and lower mechanical loss, the actuation performances of PFED10 are better than those of VHB 4910 under the same experimental conditions. As shown in Fig. R12, PFED10 achieves a maximum area strain of 253% at only 46 MV m⁻¹ with 275% biaxial pre-stretch, which is superior to most DEs developed in recent years (Fig. R13a). The 253% area strain of PFED10 corresponds to 72% thickness strain. The thickness strain of PFED10 is almost 5 times that of VHB 4910, which is close to the result that the

electromechanical sensitivity of PFED10 is 7 times higher than that of VHB 4910. Notably, our DE exhibits an ultrahigh specific energy of 225 J kg⁻¹ and a high specific power of 2245 W kg⁻¹ at 40 MV m⁻¹ and 5 Hz, outperforming all of the reported DEs (Fig. R13b).

In a word, higher dielectric constant and desirable mechanical properties enable our PFED10 to achieve larger actuation area strain and ultrahigh specific energy under lower electric fields, demonstrating better performance than VHB 4910 and even other DEs.

Fig. R12 The actuation area strains of VHB 4910 and PFED10 under different electric fields with 275% biaxial pre-stretch.

Fig. R13 Actuation performance of PFED10. a, Ashby plot summarizing maximum actuation area strains to electric fields of this work and other pre-stretched and stable DEs. b, Ashby plot summarizing specific energy to specific power of this work and other DEs. VHB 4910 represented the measured value of VHB 4910 in this work.

Comment 12. Which is the maximum achieved actuation at the dielectric breakdown when the actuator is not prestrained?

Response 12: As shown in Fig. R14, the maximum area strain of PFED10 before the dielectric breakdown is 25% when the dielectric elastomer actuator is without pre-stretch.

Fig. R14 The actuation area strains of VHB 4910 and PFED10 under different electric fields without pre-stretch.

Comment 13. Why did the authors choose to conduct a 2.75×2.75 biaxial pre-stretch?

Response 13: Pre-stretch can increase the actuation area strain of dielectric elastomers (*Science* 2000, 287, 836-839). 200%-300% biaxial pre-stretch is widely used on dielectric elastomers (*Adv. Mater.* 2008, 20, 621-625; *Adv. Funct. Mater.* 2021, 31, 2008321; *Macromol. Rapid Commun.* 2023, 44, 2300160). In order to maximize the actuation area strain, we measured the actuation area strains of dielectric elastomers at different biaxial pre-stretch ratios. As shown in Fig. R15, PFED10 achieved the maximum actuation area strains of 253% at $275\% \times 275\%$ biaxial pre-stretch. Therefore, we chose to conduct a biaxial pre-stretch ratio of 275%.

Fig. R15 The maximum actuation area strains at different biaxial pre-stretch ratios.

Comment 14. A maximum area strain of 253% at $46 \text{ V } \mu\text{m}^{-1}$ was achieved for the new material, which was almost 14 times larger than VHB 4910. Given that the sensitivity factor was only 7 times higher, an explanation is needed for this improvement.

Response 14: According to the equation 1 in Comment 11, the electromechanical sensitivity (β),

defined as the ratio of dielectric constant to Young's modulus, is proportional to the thickness strain for small strain. In order to compare with the predicted value from electromechanical sensitivity, the area strain needs to be converted to the thickness strain using equations 5 in Comment 11.

In our work, PFED10 achieved a maximum area strain of 253% at 46 MV m^{-1} , equal to a thickness strain of 72%. VHB 4910 achieved a maximum area strain of 18% at 46 MV m^{-1} , equal to a thickness strain of 15%. For thickness strain, PFED10 is about 5 times that of VHB 4910, which is close to the result that the electromechanical sensitivity of PFED10 is 7 times higher than that of VHB 4910. The definition of electromechanical sensitivity is based on the assumptions of small strains and linear elasticity, which limits its accuracy, and the nonlinear of materials, the viscoelasticity of materials, and experimental conditions also affect the deviation between actual actuation performance and electromechanical sensitivity (*Macromol. Rapid Commun.* 2009, 31, 10-36). Deviations between actual actuation performance and electromechanical sensitivity are common (*J. Polym. Sci. B Polym. Phys.* 2013, 51, 197-206; *Adv. Funct. Mater.* 2015, 25, 2467-2475; *Polym. Chem.*, 2016, 7, 2709-2719; *J. Appl. Polym. Sci.* 2022, 139, 52261). In general, the actuation performance of PFED10 is basically consistent with electromechanical sensitivity.

Comment 15. Figure 3e. Why is there such a huge difference in the actuation of films with different thicknesses?

Response 15: When the thickness of the dielectric elastomer film is extremely thin, the increased importance of inhomogeneities and film defects will have a great influence on the actuation performance that causes localized areas of high electric field and stress and results in premature breakdown (*Macromol. Rapid Commun.* 2009, 31, 10-36; *Smart Mater. Struct.* 2016, 25, 075018; *Proc. SPIE* 2015, 9430, 94301D-14). As a result, inhomogeneities and film defects are the main reasons for the different actuation performances of films with different thicknesses. The difference of actuation area strain of film with different thicknesses can also be observed in some literature related to dielectric elastomers (*Chem. Eng. J.* 2021, 405, 126634; *Macromol. Rapid Commun.* 2019, 40, 1900205). For example, Chen et al. reached a max actuation area strain of 184% with 0.2-mm film, while the max actuation area strain decreased to 93% with 13- μm film (*Chem. Eng. J.* 2021, 405, 126634). Although the actuation area strain is different for different thickness films, to the best of our knowledge, PFED10 film achieves the largest actuation area strain (141%) to date under low voltage ($< 700 \text{ V}$).

Comment 16. Cyclic test for the unrestrained membrane should also be given.

Response 16: Following the suggestion of the reviewer, we have added the cyclic test for the unrestrained membrane. As shown in Fig. R16, the unrestrained PFED10 was successfully actuated for 10000 cycles at 10.8 MV m^{-1} and 1 Hz, indicating that it exhibited excellent cycle stability for reliable operation.

We have added Fig. R16 to the Supporting Information as **Supplementary Fig. 23** and made the corresponding description in the manuscript.

Fig. R16 Cyclic actuation of PFED10 at 10.8 MV m^{-1} and 1 Hz for 10000 cycles without pre-stretch.

[Revised manuscript]

“Notably, PFED10 exhibited excellent cycle stability for reliable operation, regardless of the application of pre-stretch (Fig. 3g and Supplementary Fig. 23). As shown in Fig. 3g, the pre-stretched PFED10 was successfully actuated for 10000 cycles under a large actuation area strain of 120% at 1 Hz.”

Comment 17. Correct $V \mu\text{m}^{-1}$ to standard units.

Response 17: As suggested, in order to correct the unit of electric field to the standard, we consulted extensive literature, and found that MV m^{-1} has been widely used to express the unit of electric field in the literature related to dielectric elastomers. For example, MV m^{-1} was adopted in *Science* 2000, 287, 836-839; *Adv. Mater.* 2006, 18, 887-891; *Nat. Commun.* 2021, 12, 4517, and so on. Therefore, we selected MV m^{-1} as the unit of electric field, which is relatively standard.

We have revised all the units of electric field in the manuscript from $\text{V } \mu\text{m}^{-1}$ to MV m^{-1} .

[Revised manuscript]

“However, DEAs usually require extremely high driving electric fields ($> 100 \text{ MV m}^{-1}$) to achieve high performances due to the low dielectric constant and high stiffness of dielectric elastomers (DEs).”

“Our new DE exhibits a maximum area strain of 253% at a low driving electric field of 46 MV m^{-1} . Notably, it achieves an ultrahigh specific energy (mass energy density) of 225 J kg^{-1} at only 40 MV m^{-1} , around 6 times higher than natural muscle and twice higher than the state-of-the-art DE, and the corresponding specific power (mass power density) is 2245 W kg^{-1} .”

“As shown in Fig. 1c, our DE achieves a maximum actuation area strain of 253% at only 46 MV m⁻¹, which is superior to most DEs developed in recent years^{2,14,18-25}.”

...

Comment 18. Finally, the English language could be refined for greater clarity and coherence.

Response 18: Thanks for your comments. We have double-checked the language and corrected the mistakes. The revised manuscript is as follows.

[Revised manuscript]

“Here, we introduce polar fluorinated groups and nanodomains aggregated by long alkyl side chains into DE design, endowing DE with a high dielectric constant and desirable modulus, simultaneously achieving large actuation strain and high energy density under low driving electric fields.”

“Extensive efforts have been devoted to improving actuation performance of DEs at low driving electric fields, including bottlebrush elastomer^{15,16}, poly[styrene-b-(ethylene-co-butylene)-b-styrene] (SEBS) triblock copolymer mixed with oil¹⁷, pentablock copolymer¹⁸, and polyacrylate with optimized crosslinking network¹⁹.”

“With the combination of high dielectric constants and desirable mechanical properties, our dielectric elastomer is expected to display large actuation strain and high energy density under low electric fields.”

...

Response to Reviewer #2

Overview. *In “A large-strain and ultrahigh energy density dielectric elastomer for fast moving soft robot”, Feng et al. report a new elastomeric material that enables high specific energies at relatively low electric fields. The authors extensively characterize the mechanical and the actuation properties of the material. The presented material far exceeds previously reported values of specific energy. These extraordinary properties enable lifting large loads and fast robots.*

The presented work provides a clear advance for dielectric elastomers, so it will be highly interesting to the readership of Nature Communications. The work is well described and the authors provide all important metrics that are relevant for the dielectric elastomer community. I thus have only few comments, which I recommend the authors to address, before accepting the article:

Response: We thank the reviewer for the positive comments.

Comment 1. It is very interesting that the specific energy reported here for VHB, which has been a standard material for DEAs for a long time, is comparable to the specific energy of PHDE (ref. 6), a material that was recently reported in Science to provide a large increase in specific energy compared to other materials. I am wondering how much the experimental conditions influence the measured performance metrics and how comparable values from different articles really are. Please discuss.

Response 1: The specific energy is measured based on a pure-shear DEA (*Science* 2018, 359, 61-65; *Science* 2022, 377, 228-232), which was fabricated by applying a fixed pre-stretch in one plane direction and loading in the perpendicular plane direction. When voltage was applied, the expansion of DEA in the lateral direction was constrained, while it could freely expand in the load direction to produce the linear actuation. The specific energy (E) was calculated according to the increase of load potential energy during the film contraction, as shown in the following equation $E = \frac{m_L gh}{m_a}$,

where m_L is the mass of load, m_a is the mass of active area, h is the increased height of load and is equal to the linear actuation displacement of DEA. According to the equation, the specific energy is related to the mass of load, the mass of active area, and the linear actuation displacement of DEA. As a result, the experimental conditions including the mass of load, driving electric field, driving frequency, active area, and thickness of DE film can be adjusted to obtain the maximum specific energy.

As shown in Fig. R17a, b, with the increase of electric field, the linear actuation strain increases, leading to a higher specific energy. Due to the viscoelasticity of DE, the driving frequency will affect the linear actuation strain of DEA. As shown in Fig. 17c, d, DEA produced large linear actuation strains to achieve high specific energy at low frequencies. Moreover, the mass of load has a great influence on specific energy (Fig. 17a, b). The specific energy will improve with the increase of the mass of load in a certain range. In addition, the active area and thickness of DE film will affect the mass of active area. As a result, in our experiment, we select a large mass of load (120 g), a high driving electric field (40 MV m⁻¹), a low frequency (0.5~5 Hz), a thin film (initial 0.5 mm), and an appropriate active area (~ 4 cm×2.5 cm) for testing conditions to optimize the maximum specific energy.

Thanks to our DE's ability to load a heavy mass of 120 g and achieve a large linear actuation strain of 50%, our PFED10 can achieve an ultrahigh specific energy (at 40 MV m⁻¹ and 0.5 Hz). In Shi et al.'s work, the PHDE film can be loaded with 100 g but only achieves a linear actuation strain of 25% at 62.5 MV m⁻¹ and 0.5 Hz, limiting its specific energy (*Science* 2022, 377, 228-232). In our test, VHB 4910 can achieve a linear actuation strain of 16% with a load of 120 g (at 40 MV m⁻¹ and 0.5 Hz), which is comparable to the specific energy of PHDE. Our experimental conditions are similar to Shi et al.'s work, but the specific energy of PFED10 and PHDE are different, mainly because of the difference in linear actuation strain of DEA and the mass of load. Therefore, we believe that the specific energy will be affected by the experimental conditions to some extent, but more depends on the actuation performance and mechanical properties of DE itself, i.e., the linear actuation strain and the mass that the film can load.

The values of maximum specific energy from different articles are comparable (*Science* 2022, 377, 228-232; *Proc. Natl. Acad. Sci. USA* 2019, 116, 2476-2481). According to the above formula, the physical meaning of specific energy is the work done per unit mass of active material. The more

mass a DE can load and the more linear actuation strain it can achieve, the higher its specific energy. The values of specific energy can basically reflect the mechanical properties and actuation performance of DE itself.

Fig. R17 Specific energy of PFED10. a, Specific energy of VHB 4910 and PFED10 films at different electric fields with a 60 g load. The frequency was fixed at 0.5 Hz. b, Specific energy of VHB 4910 and PFED10 films at different electric fields with a 120 g load. The frequency was fixed at 0.5 Hz. c, Specific energy of VHB 4910 and PFED10 films at different frequencies with a 60 g load. The driving electric field was fixed at 20 MV m⁻¹. d, Specific energy of VHB 4910 and PFED10 films at different frequencies with a 120 g load. The driving electric field was fixed at 40 MV m⁻¹.

Comment 2. The authors report that their material is self-healing. How much does the actuation performance change after self-healing (e.g., maximum actuation strain, breakdown field)?

Response 2: For maximum actuation strain, as shown in Fig. R18a, the initial PFED10 elastomer exhibited a maximum actuation area strain of 25% at 13 MV m⁻¹ without pre-stretch. When a 125% biaxial pre-stretch was applied, the initial PFED10 elastomer achieved a maximum actuation area strain of 30.7% at 17.1 MV m⁻¹. Then we cut the initial PFED10 elastomer open with a knife and left them self-healing in water for 3 hours. As shown in Fig. R18b, the healed PFED10 elastomer reached a maximum actuation area strain of 12.1% at 11.5 MV m⁻¹ without pre-stretch and achieved a maximum actuation area strain of 17.4% at 14.5 MV m⁻¹ with 125% biaxial pre-stretch. In a word, the maximum actuation area strain was maintained at about 50% of the initial value after self-healing.

For the breakdown field, we have added additional experiments to measure the electrical

breakdown strength before and after self-healing without pre-stretch. The electrical breakdown strength of dielectric elastomer was measured by a high voltage tester (BDJC-50kV, Beijing, Beiguang). The dielectric elastomers were placed between two 10-mm-diameter parallel electrodes, which were soaked in the silicone oil at room temperature, and then a DC voltage ramp of 500 V s^{-1} was applied to the electrodes until breakdown. We tested ten samples for initial and healed PFED10 and further fitted the breakdown strength data to the Weibull distribution function:

$$F(E_B) = 1 - \exp\left(-\left(\frac{E_B}{\eta}\right)^\beta\right) \quad (1)$$

where $F(E_B)$ is the cumulative probability of electric failure, E_B is the electrical breakdown strength measured for each sample in the experiments, η is the characteristic breakdown strength which is determined from the distribution at which 63.2% of the films have broken down electrically, and β is the shape parameter that evaluates the scatter of data.

As shown in Figs R19 and R20, the electrical breakdown strength of initial PFED10 samples and healed PFED10 samples are 23.9 MV m^{-1} and 21.4 MV m^{-1} , respectively. The breakdown strength was maintained at about 89% of the initial value after self-healing.

In general, upon mechanical scratches, the DEA based on PFED10 can self-heal and maintain the basic actuation performance. We have added Figs. R19 and R20 to the Supporting Information as Supplementary Fig. 24 and made the corresponding description in the manuscript.

Fig. R18 Self-healing property of DEA based on PFED10. a, The actuation area strains of PFED10 before and after self-healing in water for 3 h without pre-stretch. b, The actuation area strains of PFED10 before and after self-healing in water for 3 h with 125% biaxial pre-stretch.

Fig. R19 Electrical breakdown strength of initial PEFD10 without pre-stretch.

Fig. R20 Electrical breakdown strength of healed PEFD10 without pre-stretch.

[Revised manuscript]

“In addition, the PFED10-based DEA has self-healing property. Upon mechanical scratches, the DEA can self-heal and maintain the basic actuation performance (Supplementary Figs. 24, 25, and 26).”

“Dielectric properties characterization: Dielectric properties as a function of frequency were measured by a Broadband Dielectric Spectroscopy (Alpha-T, Novocontrol Technologies GmbH & Co. KG) at room temperature. The samples were cut into circles with a diameter of 10 mm and a thickness of 1 mm, and then placed between parallel electrodes. Frequency sweep range was from 10 Hz to 10⁶ Hz. The conductivity as a function of temperature was similarly tested from -20 °C to 100 °C. The electrical breakdown strength of dielectric elastomer was measured by a high voltage tester (BDJC-50kV, Beijing, Beiguang). The dielectric elastomers were placed between two 10-mm-diameter parallel electrodes, which were soaked in the silicone oil at room temperature, and then a DC voltage ramp of 500 V s⁻¹ was applied to the electrodes until breakdown. We tested ten samples for initial and healed PFED10 without pre-stretch and further fitted the breakdown strength

data to the Weibull distribution function, $F(E_B)=1-\exp(-(E_B/\eta)^\beta)$, where $F(E_B)$ is the cumulative probability of electric failure, E_B is the electrical breakdown strength measured for each sample in the experiments, η is the characteristic breakdown strength which is determined from the distribution at which 63.2% of the films have broken down electrically, and β is the shape parameter that evaluates the scatter of data.”

Comment 3. Throughout the article, the authors report performance metrics normalized by the mass of the material and term it “... density” (e.g., energy density). This term is normally reserved for metrics that are normalized by their volume. For normalization by mass, the term “specific ...” is used (e.g., specific energy). Please change the terms throughout the text.

Response 3: Thanks for your comments. We have revised the corresponding expression in the manuscript.

[Revised manuscript]

“With a load of 60 g, the specific energy of PFED10 can reach up to 75 J kg^{-1} during contraction at 20 MV m^{-1} (block force $\sim 340 \text{ mN}$) while VHB 4910 is only 16 J kg^{-1} (Supplementary Fig. 27). Impressively, PFED10 achieved 50% linear strain and exhibited a maximum specific energy of 208 J kg^{-1} with a load of 120 g at 40 MV m^{-1} and 0.5 Hz (block force $\sim 594 \text{ mN}$), which was almost 3 times higher than that of VHB 4910 (71 J kg^{-1}) (Fig. 4b, c and Supplementary Fig. 28).”

“Notably, at 5 Hz, PFED10 reached an ultrahigh specific energy of 225 J kg^{-1} , which was almost 6 times higher than that of natural muscle ($0.4\text{-}40 \text{ J kg}^{-1}$) and even twice higher than the state-of-the-art dielectric elastomers⁶ ($\sim 100 \text{ J kg}^{-1}$), and the corresponding specific power was up to 2245 W kg^{-1} (Fig. 4d, e and Supplementary Video 2). In contrast, the specific energy and specific power of VHB 4910 were 78 J kg^{-1} and 784 W kg^{-1} , respectively, at 5 Hz. As the frequency increases, the specific energy and corresponding specific power of PFED10 remained at 43 J kg^{-1} and 855 W kg^{-1} at 10 Hz, and 21 J kg^{-1} and 855 W kg^{-1} at 20 Hz. The maximum specific energy and specific power of PFED10 surpass all the reported DEs (Fig. 1d).”

“Moreover, PFED10 exhibited excellent cycle stability under pure-shear linear actuation as well. PFED10 can output a high specific energy above 150 J kg^{-1} and a high specific power above 1500 W kg^{-1} for 10000 cycles (Supplementary Fig. 29).”

...

Comment 4. I recommend to also show the measured performance metrics for VHB in Figure 1c,d as they are also very high compared to most previous literature.

Response 4: Following the suggestion of the reviewer, we have marked the measured performance metrics of VHB 4910 in Fig. 1c, d, and the updated Fig. 1 is as follows.

[Revised manuscript]

Fig. 1 Polymer design and high-performance actuation. **a**, Schematic illustration of actuation mechanism and chemical structures of our high-performance DE. **b**, Variation of the stress-strain curves of DA-aggregated nanodomains physical crosslinked network and uncrosslinked network. **c**, Ashby plot summarizing maximum actuation area strains to electric fields of this work and other pre-stretched and stable DEs. **d**, Ashby plot summarizing specific energy to specific power of this work and other DEs. VHB 4910 represented the measured value of VHB 4910 in this work.

Comment 5. Some references to supplementary figures seem to be incorrect (e.g., page 7, line 137 refers to Supplementary Figs. 12 and 13, but should refer to Supplementary Figs. 14 and 15).

Response 5: Thanks for your suggestion. Supplementary Figs. 12 and 13 in the original manuscript showed the self-healing properties of the material PFED10. Supplementary Figs. 14 and 15 in the original manuscript showed the self-healing properties of DEA based on PFED10. On page 7 of the original manuscript, we described the self-healing properties of the material. Following the suggestion of the reviewer, we have double-checked the references to supplementary figures to make sure there were no incorrect references to figures in the new manuscript.

Response to Reviewer #3

Overview. *To achieve high dielectric constant and low modulus of DEs, the authors introduce polar fluorinated groups and nanodomains aggregated by long alkyl side chains into DE design, which simultaneously achieves large actuation strain and high energy density under a low driving electric field. The new DE exhibits a maximum area strain of 253% at a low driving electric field of $46 \text{ V } \mu\text{m}^{-1}$. Notably, it achieves an ultrahigh energy density of 225 J kg^{-1} at only $40 \text{ V } \mu\text{m}^{-1}$, around 66 times that of natural muscle and twice that of the state-of-the-art DE, and the corresponding power density reaches 2245 W kg^{-1} . Moreover, the DE has excellent cycle stability and good performance for soft robot application. The work reaches a high level, and the designed DE is promising for electric field actuation such as soft robot application. There are some questions that need to be addressed before publication.*

Response: We thank the reviewer for the positive comments.

Comment 1. The language should be double-checked. Some obvious mistakes are as follows:

- 1) Line 14, “endow DE with high dielectric constant and desirable modulus”
- 2) Line 98, “elastomer is expected to displays large actuation strain”
- 3) Lines 94-95, “To tune the mechanical and dielectric properties of our DE, the molar ratio of HFBA, EA 94 and DA were systematically changed”
- 4) Line 130, “good elasticity of PFED10 were arise from the nanodomains”
- 5) Lines 131-133, “Compared to PFE, the SAXD result of PFED10 131 were featured with a peak like PD”, “which indicated the consist of nanodomains and the 132 average distance d ($d = 2\pi/q_{max}$) between the nanodomains was about 3.76 nm”
- 6) Line 178, “In addition, the PFED10-based DEA have self-healing property”
- 7) ...

Response 1: Thanks for your careful comments. We have made a point-to-point revision in response to these comments.

- 1) “Here, we introduce polar fluorinated groups and nanodomains aggregated by long alkyl side chains into DE design, **endowing** DE with a high dielectric constant and desirable modulus, simultaneously achieving large actuation strain and high energy density under low driving electric **fields**.”
- 2) “With the combination of high dielectric constants and desirable mechanical properties, our dielectric elastomer is expected to **display** large actuation strain and high energy density under low electric **fields**.”
- 3) “To tune the mechanical and dielectric properties of our DE, the molar **ratios** of HFBA, EA, and DA were systematically changed (Supplementary Table 1).”
- 4) “The SAXD result confirmed **that** the strain-hardening behavior and good elasticity of PFED10 **arose** from the nanodomains aggregated by DA segments, which served as dynamic physical crosslinkers.”
- 5) “Compared to PFE, the SAXD result of PFED10 **showed a peak similar to that of PD⁴⁴, indicating the presence of nanodomains. The average distance d between nanodomains ($d = 2\pi/q_{max}$) was approximately 3.76 nm (Fig. 2g and Supplementary Fig. 19).**”
- 6) “In addition, the PFED10-based DEA **has** self-healing property.”
- 7) “Extensive efforts have been devoted to **improving** actuation performance of DEs at low driving electric **fields**.”
- 8) “To **verify** the structure of PFED10, we used wide-angle X-ray diffraction (WAXD) and small-angle X-ray diffraction (SAXD) tests (Fig. 2g and Supplementary Figs. 18 and 19).”
- 9) “Further, the actuation performances were **tested** with 2.75×2.75 biaxial pre-stretch.”

10) “Benefiting from the ultrahigh specific energy, the PFED10-based soft robot possessed good load-carrying ability.”

...

Comment 2. Confirm if the expression is correct “The stress-strain curve of PE is soft and weak or brittle” in Supplementary Fig. 2.

Response 2: Following the suggestion of the reviewer, we have revised the corresponding expression in the manuscript.

[Revised manuscript]

“Supplementary Fig. 2. Stress-strain curves of poly(2-ethylhexyl acrylate) (PE) and poly(dodecyl acrylate) (PD). According to the stress-strain curves, the mechanical property of PE is soft and weak. The mechanical property of PD is soft but brittle. The low stretchability of PD comes from the aggregation of the long side chains serving as physical crosslinkers.”

Comment 3. According to the performance of the samples, the fraction of DA is critical to the properties. Why is molar ratio of 10 the best? More discussion should be given.

Response 3: In our PFED series elastomers, DA serves two important roles for excellent actuation performances: i) enhance the elasticity and ii) achieve strain-hardening behavior. For elasticity, the more DA content, the better the elasticity of the elastomer (Fig. R21a). For strain-hardening behavior, with the increase of DA content, the strain-hardening behavior of the elastomer became more obvious (Fig. R21b). The strain-hardening behavior will greatly affect the maximum actuation area strain. As shown in Fig. R21c, d, if the material does not exhibit stress-enhanced behavior, it cannot achieve a large actuation area strain, such as PFE. With the addition of DA, the elastomers exhibited strain-hardening behavior and the maximum actuation area strain also increased rapidly. However, as shown in Fig. R21d, the maximum actuation area strain did not improve continuously with the increase of DA content, and there is a peak value. This is because when the content of DA is too high, its obvious strain-hardening behavior makes the pre-stretched polymer chain too stiff, limiting the maximum actuation area strain. In addition, the elongation at break of elastomer decreased with the increase of DA content, which affected the subsequent biaxial pre-stretch. Therefore, in the series of PFED elastomers, we selected PFEDA10 with a DA molar ratio of 10% for our study, which had good elasticity, desirable strain-hardening behavior, suitable elongation at break, and the largest actuation area strain.

To explain more clearly, we have revised the corresponding expression in the manuscript.

Fig. R21 Mechanical properties and actuation performances of copolymers. a, Cyclic stress-strain curves of PFED5, PFED10, and PFED20. b, Stress-strain curves of PFED5, PFED10, and PFED20. c, Stress-strain curves of PFE. d, The maximum actuation area strain of PFE, PFED5, PFED10, and PFED20.

[Revised manuscript]

“In addition, all PFED copolymers can achieve a large actuation strain more than 150%, far exceeding that of PFE, indicating that the strain-hardening behavior is necessary to achieve large actuation performance^{3,45} (Supplementary Fig. 6). In the series of PFED copolymers, PFED10 exhibited the largest actuation area strain. Therefore, considering the mechanical properties, dielectric properties, and actuation area strain, the PFED10 was selected for our study.”

Comment 4. For all PFED series, DE content is constant, why?

Response 4: In our polymer design, due to the large steric hindrance side chains, EA is very soft and is selected as a comonomer to lower the Young’s modulus of the copolymer (Fig. R22a). However, EA has poor elasticity and a low dielectric constant (Fig. R22b, c). In the copolymer, increasing the EA content will lead to a decrease in elasticity and dielectric constant (Fig. R23). Therefore, considering the mechanical and dielectric properties, we fixed a low molar ratio of EA relative to HFBA at 10%.

To explain more clearly, we have revised the corresponding expression in the manuscript and we have added Fig. R22c to the Supporting Information as **Supplementary Fig. 5**.

Fig. R22 Mechanical properties and dielectric properties of polymers. a, Stress-strain curves of PE. b, Cyclic stress-strain curves of PF, PE, and PD. c, Dielectric constants of PF, PE, and PD from 10 Hz to 10⁶ Hz. PF, PE, and PD represent polymerized HFBA, EA, and DA, respectively.

Fig. R23 The mechanical properties and dielectric properties of PF and PFE. a, Cyclic stress-strain curves of PF and PFE. b, Dielectric constants of PF and PFE from 10 Hz to 10⁶ Hz

[Revised manuscript]

“As shown in Fig. 2a, the polymerized HFAB (PF) was stiff, so EA was added to soften the copolymer (PFE). Due to the poor elasticity and low dielectric constant of EA, excessive addition of EA will decrease the elasticity and dielectric constant of the copolymer (Supplementary Figs. 4 and 5). Therefore, we fixed the molar ratio of EA relative to HFBA at 10%.”

REVIEWERS' COMMENTS

Reviewer #2 (Remarks to the Author):

The authors addressed all my comments to my satisfaction. I recommend publication of the article as is as it provides a clear advance in materials for dielectric elastomer actuators.

Reviewer #3 (Remarks to the Author):

My concerns are all addressed. The manuscript can be accepted for publication.

[Note from the Editor: Reviewer #3 was asked to look also over the response given to Reviewer #1] and thinks that the authors addressed well the concerns from Reviewer #1.